# Enhancing Contrastive Clustering with Negative Pair-guided Regularization

**Abhishek Kumar**                                    *abhishek.kumar.eee13@iitbhu.ac.in*
*ENET Centre, Centre for Energy and Environmental Technologies*
*VSB- Technical University of Ostrava, Ostrava, Czech Republic.*

**Anish Chakrabarty**                                 *chakrabarty.anish@gmail.com*
*Statistics and Mathematics Unit*
*Indian Statistical Institute, Kolkata, India*

**Sankha Subhra Mullick**                             *mullicksankhasubhra@gmail.com*
*Dolby Laboratories, India*

**Swagatam Das**                                      *swagatam.das@isical.ac.in*
*Electronics and Communication Sciences Unit*
*Indian Statistical Institute, Kolkata, India*

**Reviewed on OpenReview:** *https://openreview.net/forum?id=4VYzqQ4Me*

## Abstract

Contrastive Learning (CL) aims to create effective embedding for input data by minimizing the distance between positive pairs, i.e., different augmentations or views of the same sample. To avoid degeneracy, CL also employs auxiliary loss to maximize the discrepancy between negative pairs formed with views of distinct samples. As a self-supervised learning strategy, CL inherently attempts to cluster input data into natural groups. However, the often improper trade-off between the attractive and repulsive forces, respectively induced by positive and negative pairs, can lead to deformed clustering, particularly when the number of clusters $k$ is unknown. To address this, we propose NRCC, a CL-based deep clustering framework that generates cluster-friendly embeddings. NRCC repurposes Stochastic Gradient Hamiltonian Monte Carlo sampling as an approximately invariant data augmentation, to curate hard negative pairs that judiciously enhance and balance the two adversarial forces through a regularizer. By preserving the cluster structure in the CL embedding, NRCC retains local density landscapes in lower dimensions through neighborhood-conserving projections. This enables the application of mode-seeking clustering algorithms, typically hindered by high-dimensional CL feature spaces, to achieve exceptional accuracy without needing a predetermined $k$. NRCC's superiority is demonstrated across various datasets with different scales and cluster structures, outperforming 20 state-of-the-art methods.

## 1 Introduction

Contrastive learning is a self-supervised learning technique that trains a model by encouraging it to distinguish between similar and dissimilar pairs of data points, enhancing its ability to learn meaningful representations (Chen et al., 2020b). A representation can be considered accurate if two semantically similar samples lie close to each other in the feature space while the dissimilar ones reside at a distance. In other words, the learned embedding space should be geometry-aware, preserving the local neighborhood as well as distinguishing the natural groups present in the data. Hence, CL methods attempt to learn feature mapping by minimizing the distance between a positive pair. Such a pair is formed by a couple of augmented variants of the same sample and thus resides in close proximity, crowding the local neighborhood. Further, to maintain a considerable

separation between dissimilar inputs, CL methods may actively encourage maximizing the distance between a negative pair consisting of augmented views of two distinct samples. This auxiliary CL objective also safeguards the learned representation from collapsing into a degenerate solution. Specifically, the positive pairs induce an attraction while the negative pairs contribute a repulsion, resulting in a balanced interplay that sculpts an ideal embedding. However, such a crucial balance is challenging to achieve due to the unregulated repulsion induced by the negative pairs. Often, traditional CL methods show excessive hunger for negative pairs, severely affecting the computational cost. As a remedy, more recently, CL methods use only the positive pairs in a siamese-inspired architecture along with a regulated gradient flow (Grill et al., 2020). Specifically, one branch of the siamese network called *online* trains on the positive pair using typical gradient descent. The other branch, known as *target*, is updated through the moving average of the online branch parameters and, at the end of the training, acts as the feature encoder.

Deep Clustering (DC) methods address the classical unsupervised task of assigning input data to distinguishable natural groups or clusters using deep neural networks. Both DC and CL methods share a common interest in sculpting an embedding that conserves natural groups. Hence, CL often comes to the aid of DC methods (Li et al., 2020a). However, an embedding learned by a CL method may contain various irregularities (Chen & He, 2021; Huang et al., 2022). For example, while generating a negative pair, if the two considered samples belong to the same cluster or, worse, lie in a local neighborhood, then that may culminate in deformed cluster structures in the CL feature space (Chuang et al., 2020). If such an embedding is directly used for clustering, the performance is highly likely to be compromised even if the number of clusters $k$ is known. The remedies usually involve simultaneous optimization of CL and DC objectives, often with tailored clustering methods (Li et al., 2021) and always requiring the $k$ in advance (Huang et al., 2022). Moreover, the CL embedding resides in a high-enough dimensional space where applying state-of-the-art mode-seeking clustering algorithms (Jang & Jiang, 2021) that do not require prior knowledge of $k$ is infeasible (Huang et al., 2022). Given that the cluster structure has a risk of not being adequately reflected in the representation, mode-seeking clustering may also turn futile.

To address these issues regarding the applicability of CL in DC, we propose a Negative pair-based Regularizer for Contrastive Clustering (NRCC). The motivation behind NRCC is to judiciously introduce a higher repulsive force through tailored negative pairs that best preserve cluster structure in the CL embedding. There are two obstacles to this approach. First, if the negative pair-induced repulsion supersedes the positive pair-driven attraction, that may lead to undesirable fragmented clusters. On the other hand, in the absence or lack of adequate repulsion, clusters may collapse on each other. Second, if the negative pairs are generated using conventional and unregulated data augmentation techniques (Chen et al., 2020b), they suffer significant distortion if made to reside at greater distances. As such, bringing to bear a heightened repulsion comes at the price of the data density getting deformed. These factors motivate us to explore two remedial directions. First, we search for a data augmentation strategy that generates an approximately invariant (Chen et al., 2020a) view with enhanced departure, still limiting the dissimilarity from the original data distribution. Specifically, the augmentation produces a density-aware crowding of views in the vicinity of the parent sample. The resultant hard-negative pairs exert only enough repulsion to curve out finer cluster boundaries. Second, we construct a regularizer that seamlessly integrates the negative pairs in the existing CL frameworks like traditional InfoNCE (Chen et al., 2020b) or positive-pair-only siamese-inspired BYOL (Grill et al., 2020).

We consider re-purposing the Stochastic Gradient Hamiltonian Monte Carlo (SGHMC) sampling method to address the first hurdle as an augmentation strategy (Chen et al., 2014). As per our knowledge, this is the first work where the applicability of SGHMC (or in general a sampling technique) as a potential augmentation method in CL has been investigated. Originally designed as a scalable improvement over the Hamiltonian Monte Carlo (HMC) technique, SGHMC replaces the computationally expensive exact gradient calculation in HMC with an economic estimate. Further, as an iterative sampling method, SGHMC follows a Markov process to stochastically transform a seed into a sample from an intractable distribution. In our case, such a technique can instead be applied as an augmentation method by propagating toward the distribution of negative pairs with increased dissimilarity in the embedding space. Throughout the rest of this article, we establish SGHMC as a suitable augmentation strategy following approximate invariance in theory. Moreover, in practice, SGHMC allows tuning the number of iterations to control the shift of the generated view from the seed sample. These, in turn, ensure that through SGHMC, the distances between a negative pair increase

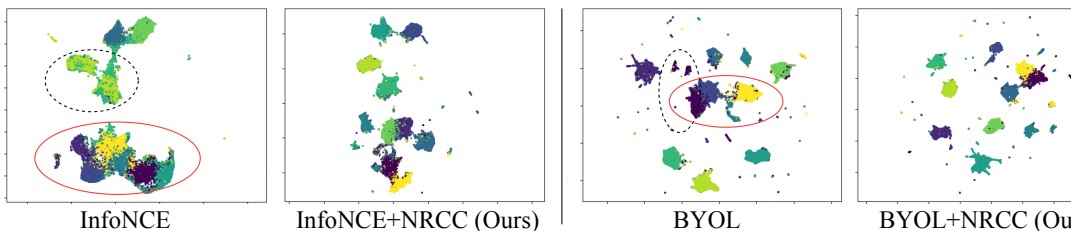

Figure 1: UMAP projections of ImageNet-10 embeddings for InfoNCE, BYOL, InfoNCE+NRCC, and BYOL+NRCC. The cluster collapse and fragmentation are respectively shown using solid red and dotted black borders. The proposed NRCC brings visible improvement in retaining cluster structure in the UMAP space compared to the corresponding non-regularized baseline CL methods. Given that UMAP preserves local neighborhoods, we conclude that NRCC also obtains a cluster-friendly embedding space.

on average while the views reside in the local neighborhood. The second challenge is solved using additional attraction and repulsion through the existing positive pairs, and SGHMC-curated negative pairs that are traded off against each other. Such a strategy can be added directly with InfoNCE as the paradigm already allows negative pairs. However, minimal architectural and algorithmic modifications are required for positive pair-only BOYL to cater to the effective usage of negative pairs.

The novelty and usefulness lie in the proficiency of NRCC at aiding CL in finding cluster-friendly embeddings. Further, such embeddings can be safely projected to lower dimensions by neighborhood preserving techniques (Van der Maaten & Hinton, 2008; McInnes et al., 2018) to allow mode-seeking clustering algorithms without requiring $k$. As a motivating example in Figure 1, we present the UMAP (McInnes et al., 2018) projections of embeddings learned by InfoNCE, BYOL, and their NRCC regularized counterparts on the ImageNet-10 (Russakovsky et al., 2015) dataset. From Figure 1, it is evident that InfoNCE suffers from both cluster collapse and fragmentation. BYOL, though improved, is still susceptible to collapse. On the other hand, NRCC, in both the cases, exhibits considerable improvement in preserving cluster structures.

Highlights from our contributions are as follows:

- We propose a new regularization framework, NRCC, that uses hard negatives and can be seamlessly coupled with the existing CL paradigms, such as InfoNCE and BYOL, to learn a clustering-friendly embedding space.

- We provide a theoretical characterization of negative pair construction that facilitates cluster-friendly representation learning. Further, we show that SGHMC sampling can be repurposed as an augmentation strategy, and the generated negative pairs satisfy approximate invariance (Chen et al., 2020a) i.e., they are hard pairs offering intensified repulsion with improved local crowding.

- NRCC relieves the user from knowing the number of clusters beforehand. Given that NRCC-enabled CL embedding preserves cluster structure, it allows for non-parametric mode-seeking clustering algorithms to be applied effectively, following a local neighborhood-preserving dimensionality reduction.

In Section 2, we review the existing works on DC and CL. Section 3 explains the NRCC regularizer, justifies both theoretically and empirically the choice of repurposing SGHMC sampling as the preferred augmentation method in NRCC, and demonstrates the promise NRCC offers when coupled with InfoNCE and BYOL. Empirical evaluation of the efficacy of NRCC with GridShift (Kumar et al., 2022) against the state-of-the-arts in Section 4, shows a 4.7% improvement in clustering accuracy by BYOL+NRCC+UMAP+GridShift on an average over eight datasets. Finally, we make concluding remarks in 5.

## 2 Related Works

### 2.1 Contrastive Learning

CL methods (Wu et al., 2018) maintain their primary focus on finding a generalized representation that can be tailored for a multitude of downstream learning tasks such as classification, clustering, etc. with minimal

additional effort. The milestone set by Chen et al. (2020b) opened a major paradigm of CL methods utilizing both positive and negative pairs. However, it was soon observed that in the case of CL, improved performance and reduced computational cost are both closely associated with finding good negative pairs (Kalantidis et al., 2020). Hence, several research directions, such as using a modified objective to prohibit similar samples from forming negative pairs (Chuang et al., 2020), theoretical and empirical characterization of generated views to find their usefulness (Tian et al., 2020), employing adversarial feedback (Hu et al., 2021), etc. were explored over the years to alleviate the problem. On another front, the work by He et al. (2020) investigated the applicability of a two-network strategy where the primary feature extractor is updated by the moving average of the parameters of an auxiliary model that goes through a traditional gradient descent. This direction later evolved into a new CL paradigm (Grill et al., 2020; Chen & He, 2021) that extended the idea of two networks through a siamese model, refined the update of the feature mapping branch with auxiliary branch parameters, and wholly discarded the need for negative pairs through judiciously controlling gradient propagation. In a less explored avenue, energy-based learning also found its usage in CL that brought back the negative pairs but attempted to limit the batch size without affecting performance (Kim & Ye, 2022). Recently, a slight deviation from generalized representation to designing a more task and data specific tailored class of CL methods gathered interest such as for computer vision problems like dehazing (Wang et al., 2024), clustering (Deng et al., 2023; Yin et al., 2023), classification (Zeng & Xie, 2021), enzyme function prediction (Yu et al., 2023), time series (Liu & Chen, 2024), graph learning (You et al., 2020; Zhang et al., 2023; Bo et al., 2024), etc. In a similar direction, we progress towards a clustering-specific CL method that effectively utilizes hard negatives to find an embedding that conserves the natural clusters present in the data.

## 2.2 Deep Clustering

Clustering performance is highly dependent on the representation of the data, i.e., how well the natural clusters are retained and expressed in the feature space. The rich embedding learned by deep neural networks found its natural usage even in the very early days (Yang et al., 2016; 2017). A primary direction for DC methods employs simultaneous optimization of multiple objectives for finding an embedding space that can be easily clustered by a specific clustering method (Guo et al., 2017; Caron et al., 2018; Asano et al., 2019; Li et al., 2020b; 2022). For example, Yang et al. (2016) proposed a deep representation learning suitable for hierarchical clustering while Caron et al. (2018) employed classification to iteratively refine weak cluster assignments. With the advent of CL, the DC research community was quick to identify the similarities between the two objectives and came up with several approaches that fuse the two for commendable clustering performances. For example, CL was used by Van Gansbeke et al. (2020) to generate reliable pseudo labels while many others (Li et al., 2021; Shen et al., 2021; Li et al., 2022; Sadeghi et al., 2022) proposed end-to-end frameworks where the CL objective was repurposed for clustering. In an allied direction, (Zhong et al., 2021; Zheng et al., 2021) graphs were employed to identify neighboring pseudo-positive samples. However, these efforts were plagued by the limitations of CL, i.e., irregularities induced by improper negative pairs, class collision (Huang et al., 2022), etc. A more recent work by Huang et al. (2022) managed to overcome the class collision problem but instead imposed a reliance on the knowledge of the number of clusters $k$ to be known in advance, limiting its practical applicability. Instead, with NRCC, we return to the initial DC strategy of employing CL to find a cluster-friendly representation that can be conserved after neighborhood-preserving dimensionality reduction. Consequently, NRCC offers flexibility for the choice of clustering methods, alleviates the impact of the curse of dimensionality, and imposes no compulsion on the prior knowledge of $k$.

## 2.3 Mode Seeking Clustering Methods

Center-based clustering methods, like $k$-means, require the knowledge of the number of clusters ($k$) in advance. Moreover, such methods demand structural and distributional constraints on the clusters. These limitations can be avoided by density-based mode-seeking clustering strategies such as MeanShift and its variants (Cheng, 1995; Jang & Jiang, 2021; Kumar et al., 2022). The MeanShift family of methods is non-parametric that does not demand prior knowledge of $k$. Depending solely on density estimates, these methods can automatically reveal natural clusters by identifying the modes of arbitrary distributions. However, MeanShift (Cheng, 1995) is not scalable due to its quadratic computational complexity with the number of samples ($O(N^2)$). This crucial shortcoming of MeanShift was addressed by MeanShift++ (Jang & Jiang, 2021) which employs a

grid structure to reduce the complexity to $O(N)$. Recently, GridShift (Kumar et al., 2022) effectively used approximated cluster centers to further achieve $\sim40\times$ speed up over MeanShift++. Moreover, GridShift demonstrated (both theoretically and empirically) that it maintains a better (or at least at per) clustering performance with MeanShift++. Based on these observations, we champion GridShift for clustering in our work. However, MeanShift++ and GridShift both fall prey to the curse of dimensionality. In both cases, the computational complexity increases exponentially with the dimension of the feature space. To fully exploit the benefits of GridShift, we ensure two properties of the CL embedding space. First, the learned embedding space correctly preserves the cluster structure. Second, the clusters in the CL embedding space can be projected on a lower-dimensional space (preferably with minimal loss of information) through neighborhood-preserving methods (Van der Maaten & Hinton, 2008; McInnes et al., 2018).

## 3 Proposed Method

### 3.1 Preliminaries

We begin with a set of $N \in \mathbb{N}_+$ samples $\{\mathbf{x}_i\}_{i\in\mathcal{I}}$ that reside in a high-dimensional space $\mathcal{X} \subseteq \mathbb{R}^d$, where $|\mathcal{I}| = N$. The high ambient dimension of $\mathcal{X}$ prevents distance-based clustering algorithms like $k$-means from performing at their true potential. Moreover, the curse of dimensionality makes the application of kernel density estimators infeasible, thereby restricting the usage of non-parametric mode-seeking clustering algorithms on such data. Hence, it becomes reasonable to project the data onto a low dimensional embedding space $\mathcal{H} \subseteq \mathbb{R}^{\tilde{d}}$, where the natural clusters are appropriately conserved or enhanced even further, $\tilde{d} \leq d$. As previously elaborated in Section 2, in the case of DC, such cluster-friendly representation space is found through a deep neural network. For example, an encoder $f : \mathcal{X} \to \mathcal{H}$ can be used to map a sample $\mathbf{x}_i$ from its native space to the corresponding latent representation $\mathbf{h}_i = f(x_i; \Omega) \in \mathcal{H}$. Here, $\Omega$ represents the set of tunable parameters that characterize $f$.

With this setup, we move on to the CL-specific case. We start with a set of possible augmentations $\mathcal{T}$ from which we randomly select two data transformations, namely $T^a$ and $T^b$. To facilitate a mini-batch learning, from $\{\mathbf{x}_i\}_{i\in\mathcal{I}}$ we randomly sample a set $\mathcal{N}$ containing $n$ data points without replacement. Now, given this mini-batch $\mathcal{N}$, a set of $n$ positive pairs is created as $\mathcal{P}_+ = \{(\mathbf{x}_i^a, \mathbf{x}_i^b)\}_{i=1}^n$, where $\mathbf{x}_i^a = T^a(\mathbf{x}_i)$ and $\mathbf{x}_i^b = T^b(\mathbf{x}_i)$ are two different views of the same data instance. As such, they share a strong semantic similarity. This aids the network in identifying the natural groups present in the data through the induced attraction while enhancing the similarity between a positive pair in the embedding space. However, naively optimizing only the positive pair-driven objective leads to a degenerate distribution supported on a single point in the embedding space. Traditionally, this is handled using an auxiliary contrastive loss that offers repulsive force by maximizing the dissimilarity between negative pairs. Formally we denote the set of negative pairs for a batch as $\mathcal{P}_- = \mathcal{P}_-^a \cup \mathcal{P}_-^b$, where $\mathcal{P}_-^a = \{(\mathbf{x}_i^a, \mathbf{x}_j^k)\}_{i,j=1,i\neq j,k\in\{a,b\}}^n$ and $\mathcal{P}_-^b = \{(\mathbf{x}_i^b, \mathbf{x}_j^k)\}_{i,j=1,i\neq j,k\in\{a,b\}}^n$. In other words, for the mini-batch $\mathcal{N}$, $\mathcal{P}_-^a$ denotes the collection of both the views $\mathbf{x}_j^a$ and $\mathbf{x}_j^b$ for each of the samples $\mathbf{x}_j$ in the set $\mathcal{N} \setminus \{\mathbf{x}_i\} = \{\mathbf{x}_1, \cdots, \mathbf{x}_{i-1}, \mathbf{x}_{i+1}, \cdots, \mathbf{x}_n\}$ paired with $\mathbf{x}_i^a$, for all $i = 1, 2, \cdots, n$ while $\mathcal{P}_-^b$ is constructed analogously. Consequently, the cardinality of $\mathcal{P}^-$ turns out to be $4n(n-1)$.

In the case of InfoNCE (Chen et al., 2020b), the CL objective is not directly optimized in the latent space learned by $f$ but in $f \circ g$, where $g : \mathcal{H} \to \mathcal{Z}$. Specifically, $g$ is realized through a dense neural network that yields $\mathbf{z}_i = g(\mathbf{h}_i)$. Assuming that $\mathbf{z}_i \in \mathcal{Z}$ is normalized, and $\tau$ is a temperature hyperparameter regulating the degree of affinity between samples, the InfoNCE loss $l^I$ can be defined as follows:

$$l^I = \sum_{i=1}^n \left( l_{i,a}^I + l_{i,b}^I \right), \text{ where} \tag{1}$$

$$l_{i,a}^I = -\frac{\langle \mathbf{z}_i^a, \mathbf{z}_i^b \rangle}{\tau} + \log \sum_{k\in\{a,b\}} \sum_{j=1,j\neq i}^n \exp\left[ \frac{\langle \mathbf{z}_i^a, \mathbf{z}_j^k \rangle}{\tau} \right], \text{ and} \tag{2}$$

$$l_{i,b}^I = -\frac{\langle \mathbf{z}_i^a, \mathbf{z}_i^b \rangle}{\tau} + \log \sum_{k\in\{a,b\}} \sum_{j=1,j\neq i}^n \exp\left[ \frac{\langle \mathbf{z}_i^b, \mathbf{z}_j^k \rangle}{\tau} \right]. \tag{3}$$

The InfoNCE loss has two major disadvantages. First, there is no provision for considering the similarity between two distinct samples $\mathbf{x}_i$ and $\mathbf{x}_j$ in a batch $\mathcal{N}$ during the construction of $\mathcal{P}_-$. If the samples $\mathbf{x}_i$ and $\mathbf{x}_j$ are from the same cluster, then treating them as a negative pair may become detrimental for convergence towards a clustering-friendly space. However, in a self-supervised setting, in the absence of labels, there is no way to be confident about the similarity between $\mathbf{x}_i$ and $\mathbf{x}_j$ in advance. Second, optimizing the loss demands a high negative pair to positive pair ratio, $2(n-1)$ to be exact (Chen et al., 2020b; Kim & Ye, 2022). However, with a high number of negative pairs comes the issue of class collision (Arora et al., 2019) that compromises the adaptability of the CL-learned feature space to clustering. A potential remedy for both of these issues comes in the form of a negative pair generation strategy that maintains a considerable distance among the negative pairs while closely adhering to the original data distribution.

An alternative way to address the issues of InfoNCE can be found through BYOL (Grill et al., 2020). Such a CL paradigm discards the negative pairs and the associated auxiliary loss altogether while escaping degeneracy. Specifically, BYOL uses a siamese-inspired architecture where the two parallel networks are called online and target, respectively. However, compared to traditional siamese architecture, the networks in BYOL have three major differences. First, target and online branches do not share a common set of weights. Second, the target branch has an additional head $q$ to map $\mathbf{z}_i \in \mathcal{Z}$ to a prediction vector. Third, only the online branch is updated using gradient descent on the loss function, while the weights of the target branch are a moving average of the online parameters. These three key modifications harmoniously ensure that both branches are not affected by the same gradient. Thus, the scenario resembles an alternating optimization that helps to avoid the degenerate collapsing. Specifically, the BYOL objective function $l^B$ can be defined as follows:

$$l^B = \sum_{i=1}^{n} l_{i,a}^B + l_{i,b}^B, \text{ where } l_{i,a}^B = ||q(\mathbf{z}_i^a) - \bar{\mathbf{z}}_i^b||_2^2 \text{ and } l_{i,b}^B = ||q(\mathbf{z}_i^b) - \bar{\mathbf{z}}_i^a||_2^2. \tag{4}$$

Note that the outputs and mappings of the target network are distinguished from their online counterparts using a bar $(\bar{\cdot})$. While BYOL directly improves over InfoNCE in general, its clustering potential may not have been fully exploited, as in the absence of counteracting repulsive forces, the embedding space may still coalesce multiple clusters (Chen & He, 2021).

## 3.2 Regularizing CL Loss for Clustering

Following the discussion in Section 3.1, it is evident that we need to focus on three primary fronts to achieve good clustering. First, maintain a trade-off between repulsive and attractive forces such that the clusters neither collapse nor fragment but remain uniformly distributed and well separated in the embedding space. Second, if similar samples form a negative pair, the repulsion force should be adjusted adaptively to protect the learning from being misguided. Third, negative pairs are the direct source of repulsion in a CL framework. Hence, ignoring them completely, as in BYOL, may not be helpful for clustering as that may lead to undesirable cluster collapse. We now proceed to detail the proposed NRCC regularizer that can be effortlessly coupled with InfoNCE or BYOL.

$$\bar{l}^I = \sum_{i=1}^{n} \bar{l}_{i,a}^I + \bar{l}_{i,b}^I, \tag{5}$$

$$\text{where } \bar{l}_{i,a}^I = l_{i,a}^I + \lambda \left[ \log \frac{\sum_{j=1}^{n} \exp(\langle \mathbf{z}_i^a, \mathbf{z}_j^b \rangle)}{\sum_{j=1}^{n} \exp(\langle \mathbf{z}_i^a, \mathbf{z}_j^c \rangle)} \right] \text{ and } \bar{l}_{i,b}^I \text{ is defined analogously,}$$

$$\bar{l}^B = \sum_{i=1}^{n} \bar{l}_{i,a}^B + \bar{l}_{i,b}^B, \tag{6}$$

$$\text{such that } \bar{l}_{i,a}^B = l_{i,a}^B + \lambda \left[ \log \frac{\sum_{j=1}^{n} \exp(\langle \mathbf{z}_i^a, \bar{\mathbf{z}}_j^b \rangle)}{\sum_{j=1}^{n} \exp(\langle \mathbf{z}_i^a, \bar{\mathbf{z}}_j^c \rangle)} \right] \text{ and } \bar{l}_{i,b}^B \text{ is defined similarly.}$$

In the above formulation, $\bar{l}^I$ and $\bar{l}^B$ are, respectively, the regularized InfoNCE and BYOL losses. This notational convention also extends to the sample-specific objectives, i.e., $\bar{l}_{i,a}^I$ is the regularized form of $l_{i,a}^I$

along with the other similar terms. We introduce a new Lagrangian hyper-parameter $\lambda$ that controls the relative weight of the regularizer. Moreover, $\mathbf{z}^c$ or in case of BYOL $\bar{\mathbf{z}}^c$ corresponds to an arbitrary augmented view of $\mathbf{x} \in \mathcal{N}$. Additionally, for InfoNCE following the canonical objective, the temperature $\tau$ can be included to scale the inner products for finer control of the attractive and repulsive forces.

To better understand how the proposed NRCC regularizer alters the attraction and repulsion in an effort to achieve better clustering, we take a closer look at these forces under the light of the gradients. For simplicity without loss of generality, we only focus on the loss gradients with respect to $\mathbf{z}_i^a$, as the same for other cases follow a similar derivation. In the case of the traditional InfoNCE loss,

$$
\begin{aligned}
\frac{\partial l^I}{\partial \mathbf{z}_i^a} &= \frac{1}{2n} \left[ \frac{\partial l_{i,a}^I}{\partial \mathbf{z}_i^a} + \frac{\partial l_{i,b}^I}{\partial \mathbf{z}_i^a} + \sum_{j=1, j\neq i}^n \sum_{k\in\{a,b\}} \frac{\partial l_{j,k}^I}{\partial \mathbf{z}_i^a} \right] \\
&= \frac{1}{2n} \left[ -\frac{2\mathbf{z}_i^b}{\tau} + \sum_{j=1, j\neq i}^n \sum_{s\in\{a,b\}} \left[ \frac{\{\exp(\langle \mathbf{z}_i^a, \mathbf{z}_j^s\rangle/\tau)\mathbf{z}_j^s\}/\tau}{\sum_{k\in\{a,b\}} \sum_{\substack{l=1 \\ l\neq i}}^n \exp(\langle \mathbf{z}_i^a, \mathbf{z}_l^k\rangle/\tau)} + \frac{\{\exp(\langle \mathbf{z}_i^a, \mathbf{z}_j^s\rangle/\tau)\mathbf{z}_j^s\}/\tau}{\sum_{k\in\{a,b\}} \sum_{\substack{l=1 \\ l\neq i}}^n \exp(\langle \mathbf{z}_l^k, \mathbf{z}_j^s\rangle/\tau)} \right] \right]. \quad (7)
\end{aligned}
$$

Let us now rephrase the expression (7) in terms of the two forces. If we denote $\frac{\partial l_-^I}{\partial \mathbf{z}_i^a}$ and $\frac{\partial l_+^I}{\partial \mathbf{z}_i^a}$ as respectively the negative gradient or attraction and positive gradient or repulsion then,

$$
\frac{\partial l^I}{\partial \mathbf{z}_i^a} = -\frac{\partial l_-^I}{\partial \mathbf{z}_i^a} + \frac{\partial l_+^I}{\partial \mathbf{z}_i^a}, \text{ such that,}
$$

$$
\frac{\partial l_-^I}{\partial \mathbf{z}_i^a} = \frac{\mathbf{z}_i^b}{n\tau}, \quad (8)
$$

$$
\frac{\partial l_+^I}{\partial \mathbf{z}_i^a} = \frac{1}{2n} \sum_{j=1, j\neq i}^n \sum_{s\in\{a,b\}} \psi_{j,s} \mathbf{z}_j^s \exp(\langle \mathbf{z}_i^a, \mathbf{z}_j^s\rangle/\tau)/\tau, \quad (9)
$$

$$
\text{where } \psi_{j,s} = \left[ \sum_{k\in\{a,b\}} \sum_{l=1, l\neq i}^n \exp(\langle \mathbf{z}_i^a, \mathbf{z}_l^k\rangle/\tau) \right]^{-1} + \left[ \sum_{k\in\{a,b\}} \sum_{l=1, l\neq i}^n \exp(\langle \mathbf{z}_l^k, \mathbf{z}_j^s\rangle/\tau) \right]^{-1}.
$$

If we now denote the modified forces for the NRCC regularized variant of InfoNCE as $\frac{\partial \bar{l}_-^I}{\partial \mathbf{z}_i^a}$ and $\frac{\partial \bar{l}_+^I}{\partial \mathbf{z}_i^a}$, then

$$
\frac{\partial \bar{l}_-^I}{\partial \mathbf{z}_i^a} = \frac{\partial l_-^I}{\partial \mathbf{z}_i^a} \left[ 1 + \lambda \frac{\exp(\langle \mathbf{z}_i^a, \mathbf{z}_i^b\rangle/\tau)}{\sum_{l=1}^n \exp(\langle \mathbf{z}_i^a, \mathbf{z}_l^b\rangle/\tau)} \right], \quad (10)
$$

$$
\frac{\partial \bar{l}_+^I}{\partial \mathbf{z}_i^a} = \frac{\partial l_+^I}{\partial \mathbf{z}_i^a} + \lambda \sum_{j=1, j\neq i}^n \left[ \frac{\mathbf{z}_j^c \exp(\langle \mathbf{z}_i^a, \mathbf{z}_j^c\rangle/\tau)/\tau}{\sum_{l=1}^n \exp(\langle \mathbf{z}_i^a, \mathbf{z}_l^c\rangle/\tau)} - \psi_j' \mathbf{z}_j^b \exp(\langle \mathbf{z}_i^a, \mathbf{z}_j^b\rangle/\tau)/\tau \right], \quad (11)
$$

$$
\text{where } \psi_j' = \left[ \sum_{l=1}^n \exp(\langle \mathbf{z}_i^a, \mathbf{z}_l^b\rangle/\tau) \right]^{-1} + \left[ \sum_{l=1}^n \exp(\langle \mathbf{z}_j^b, \mathbf{z}_l^a\rangle/\tau) \right]^{-1}.
$$

There are three key takeaways from the above discussion:

1. Comparing (8) with (10), it becomes evident that the proposed NRCC further strengthens the attraction from its non-regularized counterpart.

2. If we consider (9) and (11), then notice that there are two terms of opposite signs that are newly introduced by the regularizer. The first term on the left acts as a countermeasure against negative pairs made of semantically similar samples. In such cases, this term is likely to be larger, thereby diminishing the effective repulsion to limit the impact of an improper negative pair on the learning.

3. The second term in the right introduced by the regularizer directly enhances the repulsive force.

We can now move on to BYOL and undertake a similar study. However, unlike InfoNCE, in the case of BYOL, there are additional implementation-specific heuristics like the predictor block and stop gradient. These heuristics are important factors to consider when finding an effective position for plugging in the proposed NRCC regularizer in the BYOL architecture. For the moment, we consider a theoretically simpler setting where the additional predictor block and stop-gradient are not used while NRCC regularization is applied on $\mathcal{Z}$ similar to InfoNCE. This does not sacrifice generality as such an assumption essentially reduces $\bar{\mathbf{z}}$'s to simply $\mathbf{z}$'s. Thus, an additional multiplier $\frac{\partial \bar{\mathbf{z}}}{\partial \mathbf{z}}$ gets removed from the derivation of the canonical case making it easier to interpret. We revisit the case of coupling BYOL with NRCC, without imposing any additional simplification, in a later Section 3.6.1. Meanwhile, borrowing the notational style from InfoNCE, the analogous gradients for the canonical BYOL (although simplified) take the following forms:

$$\frac{\partial l_-^B}{\partial \mathbf{z}_i^a} = \frac{\mathbf{z}_i^b}{n} \text{ and } \frac{\partial l_+^B}{\partial \mathbf{z}_i^a} = 0. \tag{12}$$

Hence, the negative gradient $\frac{\partial l_-^B}{\partial \mathbf{z}_i^a}$ induced attraction for BYOL is similar to that of InfoNCE except for the optional $\tau$. However, in the absence of negative pairs, the repulsion force contributed by the positive gradient $\frac{\partial l_+^B}{\partial \mathbf{z}_i^a}$ becomes zero. Now, if we apply NRCC, then the forces respectively modify to,

$$\frac{\partial l_-^B}{\partial \mathbf{z}_i^a} = \frac{\mathbf{z}_i^b}{n} \left[ \frac{1}{2} + \lambda \frac{\exp(\langle \mathbf{z}_i^a, \mathbf{z}_i^b \rangle / \tau)}{\sum_{l=1}^n \exp(\langle \mathbf{z}_i^a, \mathbf{z}_l^b \rangle / \tau)} \right], \tag{13}$$

$$\frac{\partial l_+^B}{\partial \mathbf{z}_i^a} = \frac{\lambda}{2n} \sum_{j=1, j \neq i}^n \left[ \frac{\mathbf{z}_j^c \exp(\langle \mathbf{z}_i^a, \mathbf{z}_j^c \rangle / \tau) / \tau}{\sum_{l=1}^n \exp(\langle \mathbf{z}_i^a, \mathbf{z}_l^c \rangle / \tau)} - \psi_j' \mathbf{z}_j^b \exp(\langle \mathbf{z}_i^a, \mathbf{z}_j^b \rangle / \tau) / \tau \right]. \tag{14}$$

Upon closer inspection, we can see that (10) resembles the same for (13) in the sense that both enhance attraction similarly. However, the lesser additive constant, i.e., $\frac{1}{2}$ in BYOL, may slightly reduce the attractive force compared to InfoNCE. This is desirable given that BYOL, by design, is driven by a strong negative gradient; thus, to gain the benefit, the NRCC-induced attraction should be added in a regulated manner. If we move our focus to the positive gradient-driven repulsion, there also BYOL shares similarity with InfoNCE, as evident from (14) and (11). Thus, our previous discussion in the context of regularized InfoNCE still holds in the case of BYOL. In other words, the NRCC not only introduces repulsion in BYOL but also adaptively restricts the ill effects of improper negative pairs.

Till this point, we have observed the impact of NRCC in improving the clustering potential of CL methods. However, NRCC can only work at its true potential if $\mathbf{z}^c$ can produce a hard negative pair that further improves the repulsion. To this end, we take a deeper look at the standard augmentation techniques and discuss the applicability of repurposing SGHMC in the context.

### 3.3 Characterizing Data Augmentation in CL

The data samples can be deemed as instances of i.i.d. observations $X_1, X_2, \cdots, X_N \sim \mathbb{P}$, on the sample space $\mathcal{X}$. Some augmentation methods like magnification and rotation are simply affine transformations. Whereas others, such as cropping or color jitter, may not be so. Either way, semantic features from the original images mostly stay intact post-transformation. We can represent such augmentations as a group of transforms $\mathcal{T}$, acting on the sample space. As such, there exists a map $\phi : \mathcal{T} \times \mathcal{X} \to \mathcal{X}$, such that it is associative $\phi(TT^*, \mathbf{x}) = \phi(T, \phi(T^*, \mathbf{x})) = \phi(T, T^*\mathbf{x})$, for $T, T^* \in \mathcal{T}$ and given the identity $\mathbf{e} \in \mathcal{T}$, $\phi(\mathbf{e}, \mathbf{x}) = \mathbf{e}\mathbf{x} = \mathbf{x}$ (Chen et al., 2020a). In our study, we do not assume that exact invariance holds, i.e. given $X \sim \mathbb{P}$, it may so happen that $TX \neq_d X$. Under such a setup, the positive pair $(\mathbf{x}_i^a, \mathbf{x}_i^b)$ obtained by applying $T^a$ and $T^b$ independently on $\mathbf{x}_i$ may not be identically distributed. This stems from the fact that the augmentation techniques vary vastly between themselves. We emphasize that the goal of the augmentation is not achieving distributional identity. Rather, we want to simulate perturbed offshoots of the given observation that share semantic information, i.e., to attain approximate invariance (Chen et al., 2020a).

Let us now focus on the crucial task of learning to embed data from $\mathcal{X}$ onto a lower-dimensional space $\mathcal{H}$. Specifically, we employ a ResNet-type architecture (He et al., 2016) that tends to preserve the local geometry

of the input space $\mathcal{X}$ due to their capability of approximating Lipschitz maps with high precision (Oono & Suzuki, 2019), also extendable to the Bi-Lipschitz criterion. We denote a convolution operation, involving weights $w$ and bias $b$, by the notation $\mathrm{Conv}_{w,b}^{\sigma} : \mathbb{R}^{d \times C} \to \mathbb{R}^{d \times C'}$; where $\sigma$ is the activation function, applied componentwise. The input and resultant channel sizes are given as $C, C' \in \mathbb{N}_+$. Similarly, the notation used for a dense layer is $\mathrm{FC}_{W,b}^{\sigma}(\cdot) := \sigma(W \mathrm{vec}(\cdot) + b)$, a function mapping $\mathbb{R}^{d \times C'}$ to $\mathbb{R}^{\tilde{d}}$. Also, let $P : \mathbb{R}^d \to \mathbb{R}^{d \times C}$ be a padding operation, responsible for aligning channels. Thus, in an encoder network with $M$ residual blocks and each block having a depth $L$, the desired embedding becomes:

$$\Psi_{\Omega}^{\sigma} : \mathbb{R}^d \to \mathbb{R}^{\tilde{d}}, \text{ where } \Psi_{\Omega}^{\sigma} := \left[ \mathrm{FC}_{W,b}^{\mathrm{Id}} \circ (\mathrm{Conv}_{w_M, b_M}^{\sigma} + \mathrm{Id}) \circ \cdots \cdots \circ (\mathrm{Conv}_{w_1, b_1}^{\sigma} + \mathrm{Id}) \circ P \right], \qquad (15)$$

where $\mathrm{Conv}_{w_i, b_i}^{\sigma} = \mathrm{Conv}_{w_i^L, b_i^L}^{\sigma} \circ \cdots \circ \mathrm{Conv}_{w_i^1, b_i^1}^{\sigma}$. Also, $\Omega = (\{w_i^j\}_{i,j=1}^{M,L}, \{b_i^j\}_{i,j=1}^{M,L}, W, b)$ denotes the parameter space and Id are the additive identities in subsequent spaces. With this setup, let us now consider those networks that satisfy $\max_{i,j} \|w_i^j\|_{\infty} \vee \|b_i^j\|_{\infty} \leq B^c$ and $\|W\|_{\infty} \vee \|b\|_{\infty} \leq B^f$, given constants $B^c, B^f > 0$. These are our preferred candidates that induce the map $f$. By definition, $\mathbf{h}_i^k = \Psi_{\Omega}^{\sigma}(\mathbf{x}_i^k)$, $k \in \{a, b\}$. Observe that, $\mathbf{h}_i^k$'s also hail from non-identical distributions. We first show that given a desirable augmentation technique, the realized discrepancy between the embedded observation and its augmented counterpart remains bounded in expectation. This points towards a statistical characterization of augmentations that regulate the fluctuations in estimates. It also testifies that simulated positive pairs indeed crowd the local neighborhood.

Firstly, let us *assume* that the augmentation $T$ follows a probability distribution $\mathbb{Q}$ such that the output of the embedding $(X, T) \mapsto \Psi_{\Omega}^{\sigma}(TX)$ belongs to $L^2(\mathbb{P} \times \mathbb{Q})$, i.e. it has finite second moment. Let $\hat{\mathbb{P}}_N$ be the plug-in estimator based on the realizations of $X_1, X_2, \cdots, X_N$ given by $\hat{\mathbb{P}}_N := \frac{1}{N} \sum_{i \in \mathcal{I}} \delta_{X_i}$. Also let $Y_1, Y_2, \cdots, Y_N$ be a sample drawn from the $\mathbb{Q}$-augmented distribution, for example, i.i.d. copies of $\mathbf{x}^a$. Based on the same, construct $\hat{\mathbb{P}}_{T,N}$, the plug-in estimator. Given two non-negative real sequences $\{a_n\}_{n \in \mathbb{N}}$ and $\{b_n\}_{n \in \mathbb{N}}$, we denote the suppression of the universal constant $D > 0$, such that $\limsup_{n \to \infty} \frac{a_n}{b_n} \leq D$, by $a_n \lesssim b_n$.

**Proposition 3.1.** *Given that $\mathbb{P}$ is supported on a compact domain in $\mathbb{R}^d$, $d \geq 2$, there exists a deterministic error $\gamma$, obtainable using a linear program, such that for $\pi > 0$*

$$\|\mathbb{E}_{X,T} \Psi_{\Omega}^{\sigma}(\hat{\mathbb{P}}_{T,N}) - \mathbb{E}_X \Psi_{\Omega}^{\sigma}(\mathbb{P})\|_2 - \gamma \lesssim M^{-\frac{\pi}{d}} \vee d^2 N^{-\frac{1}{d}},$$

*where $M$ is the number of residual blocks present in the embedding network and the activations $\sigma$ is taken as ReLU. The transformation $\Psi_{\Omega}^{\sigma}(\mathbb{P})$ should be understood as the push-forward of $\mathbb{P}$ by $\Psi_{\Omega}^{\sigma}$.*

*Proof.* Please find the proof in Appendix B.1. $\qquad \square$

Proposition 3.1 indicates that, on average, observations obtained through a suitable augmentation technique would not deviate significantly from the original inputs in the embedding space. The worst-case discrepancy is bounded by the solution to a linear program (Sriperumbudur et al., 2012), making it finite, easily determined, and controllable. As a result, the likelihood of simulated negative pairs lying drastically farther away and causing irregular repulsion, leading to cluster disintegration, is reduced. We now focus on SGHMC, intending to repurpose it to fit the desirability standards for CL-based clustering.

### 3.4 Desirability of SGHMC as an Augmentation Technique in CL-based Clustering

In an effort to generate $\mathbf{x}_i^c$, a better-augmented view of a sample $\mathbf{x}_i \in \mathcal{N}$, that is more useful for constructing clustering-friendly negative pairs tailor-made for NRCC, we start by listing down two desirable criteria. First, compared to $\mathbf{x}_i^a$ and $\mathbf{x}_i^b$, the new augmented sample $\mathbf{x}_i^c$ should lie significantly further from $\mathbf{x}_i$ in the feature space. Second, $\mathbf{x}_i^c$ should resemble $\mathbf{x}_i$ in terms of their generating law supported on the embedding space. We initialize with a probability distribution over pairwise distances between images. Our methodology for defining the same focuses on producing a replicate that exhibits an embedding located within the local neighborhood of embedded $\mathbf{x}_i$. It is given as follows:

$$\mathbb{P}_{\Omega}(X) = \left[ 1 + \mathrm{sim}\left( g(f(X)), g(f(\mathbf{x}_i)) \right)^2 \right]^{-1}, \qquad (16)$$

where $\text{sim}(\mathbf{z}, \mathbf{z}')$ signifies a measure of similarity between $\mathbf{z}$ and $\mathbf{z}'$, e.g. the inner product. Moreover, $\Omega$ can be deemed an instance of a random variable characterizing the distribution $\mathbb{P}_\Omega$. In our discussion (see Section 3.3), $\Omega$ emerges as the set of parameters representing the encoder $f$. Now, the usage of a scalable iterative sampling method like SGHMC becomes necessary since sampling directly from $\mathbb{P}_\Omega$ becomes impracticable. We start with a randomly selected initial seed $\mathbf{s}_0 = \mathbf{x}$ from $\{\mathbf{x}_i^a\}_{i=1}^n \cup \{\mathbf{x}_i^b\}_{i=1}^n$. Given the number of iterations as $\zeta$, we also sample $\{\mathbf{r}_j\}_{j=0}^\zeta \overset{iid}{\sim} \mathcal{N}(0, I_{\tilde{d}})$. This in turn serves the initialization $\mathbf{p}_0 = \mathbf{r}_0$. Over the iterations $j = 0, 1, \cdots, \zeta$; we alternatively update $\mathbf{p}$ and $\mathbf{s}$ following the formulae:

$$\text{Update } \mathbf{p}: \ \mathbf{p}_{j+1} = (1 - \delta_1)\mathbf{p}_j - \delta_2 \nabla \mathbb{P}_\Omega(X_j)|_{X_j=\mathbf{s}_j} + \delta_3 \mathbf{r}_{j+1}, \tag{17}$$

$$\text{Update } \mathbf{s}: \ \mathbf{s}_{j+1} = \mathbf{s}_j + \delta_2 \mathbf{p}_j, \tag{18}$$

where $\delta_1, \delta_2, \delta_3$ are control parameters of SGHMC. At completion of $\zeta$ iterations, we obtain $\mathbf{s}^* = \mathbf{s}_\zeta$ and return the same as the augmented view $\mathbf{x}_i^c$. Note that our expression for the SGHMC algorithm is a reparametrized discretization of the underdamped Langevin dynamics. The SGHMC method is further detailed as the following Algorithm 1 for ease of implementation.

**Algorithm 1:** Augmented view generation with SGHMC.

---

**Input:** A sample $\mathbf{x}_i \in \{\mathbf{x}_i\}_{i=1}^n$. A set of $2(n-2)$ augmented samples $\{\mathbf{x}_i^a\}_{i=1}^n \cup \{\mathbf{x}_i^b\}_{i=1}^n$, control parameters $\delta_1$, $\delta_2$, and $\delta_3$, number of maximum iterations $\zeta$.
**Output:** The generated sample $\mathbf{s}^*$ returned as the augmented view $\mathbf{x}_i^c$.

---

1 Take a random seed $\mathbf{s}_0 = \mathbf{x}$ where $\mathbf{x} \in \{\mathbf{x}_i^a\}_{i=1}^n \cup \{\mathbf{x}_i^b\}_{i=1}^n$.
2 Further, sample a set $\{\mathbf{r}_j\}_{j=0}^\zeta$ where $\mathbf{r}_j \overset{iid}{\sim} \mathcal{N}(0, I_{\tilde{d}})$ for all $j \in \{0, 1, \cdots, \zeta\}$ and set $\mathbf{p}_0 = \mathbf{r}_0$.
3 **for** $j \in \{0, 1, \cdots, \zeta\}$ **do**
4      Update $\mathbf{p}$: $\mathbf{p}_{j+1} = (1 - \delta_1)\mathbf{p}_j - \delta_2 \nabla \mathbb{P}_\Omega(X_j)|_{X_j=\mathbf{s}_j} + \delta_3 \mathbf{r}_{j+1}$
5      Update $\mathbf{s}$: $\mathbf{s}_{j+1} = \mathbf{s}_j + \delta_2 \mathbf{p}_j$.
6 **end**
7 Return $\mathbf{s}^* = \mathbf{s}_\zeta$ as the augmented sample $\mathbf{x}_i^c$.

---

The second criterion can be satisfied by showing that SGHMC acts as an approximately invariant augmentation technique and hence is desirable for CL-based clustering methods. Let us start by defining $\mathbb{P}_\Omega^n = \frac{1}{n} \sum_{i=1}^n \mathbb{P}_\Omega(\mathbf{x}_i)$, based on observations $\{\mathbf{x}_1, \mathbf{x}_2, \cdots, \mathbf{x}_n\}$ in a batch $\mathcal{N}$. To establish a statistical characterization, we first impose mild regularity conditions, standard in related literature (Raginsky et al., 2017), on the underlying distribution.

**Assumption 3.2** (Regularity of $\mathbb{P}_\Omega$). $\mathbb{P}_\Omega$ is bounded and continuously differentiable, having bounded gradients such that for any given $\Omega$, there exists $\Delta > 0$ satisfying

$$||\nabla \mathbb{P}_\Omega(\mathbf{x}) - \nabla \mathbb{P}_\Omega(\mathbf{y})|| \leq \Delta ||\mathbf{x} - \mathbf{y}||, \quad \forall \, \mathbf{x}, \mathbf{y} \in \mathcal{X}.$$

**Assumption 3.3** (Dissipativity of gradients). Given $\mathbf{x} \in \mathcal{X}$, there exists $m > 0$ and $v \geq 0$ such that

$$\langle \mathbf{x}, \nabla \mathbb{P}_\Omega(\mathbf{x}) \rangle \geq m ||\mathbf{x}||^2 - v.$$

**Assumption 3.4** (Variance control of gradient estimates). Given realizations of replicates $\Omega_1, \cdots, \Omega_n$; the function $\mathcal{G} := \mathcal{G}(\mathbf{x}, U_\Omega) = \frac{1}{n} \sum_{j=1}^n \nabla \mathbb{P}_{\Omega_{I_j}}(\mathbf{x})$ (also bounded and Lipschitz continuous in $\mathbf{x}$ due to Assumption 3.2) satisfies

$$\mathbb{E}||\mathcal{G}(\mathbf{x}, U_\Omega) - \nabla \frac{1}{n} \sum_{i=1}^n \mathbb{P}_{\Omega_i}(\mathbf{x})||^2 \lesssim ||\mathbf{x}||^2,$$

where $U_\Omega = (\Omega_{I_1}, \cdots, \Omega_{I_n})$ given that $\{I_1, \cdots, I_n\}$ is a random permutation of $\{1, \cdots, n\}$.

**Assumption 3.5.** The law corresponding to the initial state $(\mathbf{s}_0, \mathbf{p}_0)$, say $\mu_0$, satisfies

$$\int e^{\mathcal{L}(\mathbf{s}, \mathbf{p})} d\mu_0(\mathbf{s}, \mathbf{p}) < \infty,$$

where $\mathcal{L}(\cdot, \cdot)$ is the Lyapunov function.

For a detailed commentary on the significance of the assumptions, we refer to Section 4 of Raginsky et al. (2017). Note that, under assumptions 3.2-3.5, the Markov process $(\mathbf{s}_j, \mathbf{p}_j)_{j \geq 0}$ turns out to be ergodic following a unique stationary joint distribution $\Pi_\Omega(d\mathbf{s}, d\mathbf{p})$ (Gao et al., 2022). This is what imposes the near-perfect invariance to augmented outputs $\mathbf{x}_i^c$ using SGHMC. To avoid complications due to notations, we denote the $\mathbf{s}$-marginal of $\Pi_\Omega$ as $\pi_\Omega(d\mathbf{s})$. Below, we provide an exact statement that mathematically formalizes this intuition.

**Theorem 3.6** (Raginsky et al. (2017)). *Given that Assumptions 3.2-3.5 are satisfied, there exists positive constants $\{c_i\}_{i=1}^{3}$, such that for $\varepsilon > 0$ we have the 2-Wasserstein distance $\mathcal{W}_2(\mathcal{D}(\mathbf{s}_j), \pi_\Omega) \leq \varepsilon$, whenever $c_1 \delta_2^{\frac{1}{4}} + c_2 \leq \varepsilon$ and $j \geq \mathcal{O}\left(\frac{1}{\varepsilon^4} \log\left(\frac{c_3}{\varepsilon}\right)\right)$. Also, the associated excess risk turns out to be $\leq \delta' \log\left(\frac{e\Delta}{m}\left(\frac{v}{2\delta'} + 1\right)\right) + \mathcal{O}(\frac{1}{n})$, where $\delta' = \frac{d\delta_3^2}{4\delta_1}$. Here, $\mathcal{D}(\cdot)$ denotes the distribution of an underlying random variable.*

*Proof.* Please find the proof in Appendix B.2. $\qquad\square$

From Theorem 3.6, we can conclude that after a certain point in time, the augmented views iteratively obtained using SGHMC possess a distribution that lies arbitrarily close to the stationary benchmark, dependent on the target. The approximate global minimizer thus obtained, i.e. $\mathbf{s}^*$, also holds the same property. Since the 1-Wasserstein distance $\mathcal{W}_1$ is upper bounded by $\mathcal{W}_2$, the result obtained in Theorem 3.6 can also be extended to the $\mathcal{W}_1$ metric. We have previously observed the efficacy of general augmentation procedures under $\mathcal{W}_1$ (see Appendix B). This result gives us an immediate guarantee that SGHMC achieves approximate invariance in the augmented samples.

## 3.5 Revisiting Negative Pairs in NRCC

Based on the discussion in Section 3.2, it is evident that the amount of repulsion induced by NRCC can be further amplified directly by placing the negative pairs farther away in the embedding space. However, during implementation, $\mathbf{z}$'s are always normalized as $\mathbf{z}/||\mathbf{z}||_2$, in turn following a *projected* (or *offset*) distribution supported on a hyper-sphere (Mardia & Jupp (2009), Chapter 9). As such, the points being restricted on the surface, pushing two clusters wide apart brings one or both of them closer to some other cluster. This situation can only be avoided if the augmented views are aware of the density landscape and crowd the local neighborhood rather than spreading out uniformly. As per our findings in Sections 3.3-3.4, we can argue that SGHMC satisfies this owing to its approximate invariance. To further empirically validate this claim, we undertake a comparative study covering both the qualitative and the quantitative aspects. The qualitative study in Figure 2 demonstrates that the outputs generated by a single step of SGHMC are indeed visually distinct from their original counterparts, especially in contrast to the four popular data augmentations. This is quantitatively supported by Table 1, where, given 1000 ImageNet-10 samples, the mean Euclidean distance between a negative pair in the proposed BYOL+NRCC embedding space is higher for a single update of SGHMC. Thus, we can conclude that SGHMC, when used in NRCC, can produce negative pairs that simultaneously generate higher repulsion and preserve the cluster structure in the embedding space.

Table 1: Mean and Standard Deviation of Euclidean distance in the proposed BYOL+NRCC embedding space between 1000 ImageNet-10 samples and their corresponding transformed views obtained by five augmentations namely, Random color jitters, Random gray-scale, Random horizontal rotation, Random crop with resizing, and one step of SGHMC (ours).

| Data Transformation | Distance between negative pair in embedding space (Mean±Standard Deviation) |
|---|---|
| Random color jitters | 0.0038±0.0009 |
| Random gray-scale | 0.0038±0.0009 |
| Random horizontal rotation | 0.0038±0.0019 |
| Random crop with resizing | 0.0011±0.0010 |
| SGHMC (Ours) | **0.0112 ± 0.0060** |

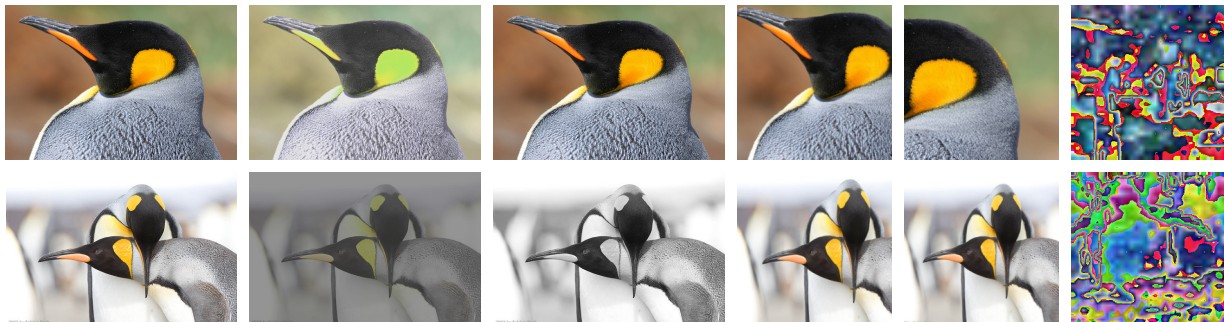

Figure 2: Augmentation outcomes corresponding to image inputs (from left): Original, Random color jitters, Random gray-scale, Random horizontal rotation, Random crop with resizing, and single step of SGHMC (ours). Compared to the other four augmentation techniques, SGHMC produces visibly distinct outputs that translate to a larger distance in the embedding space for a negative pair.

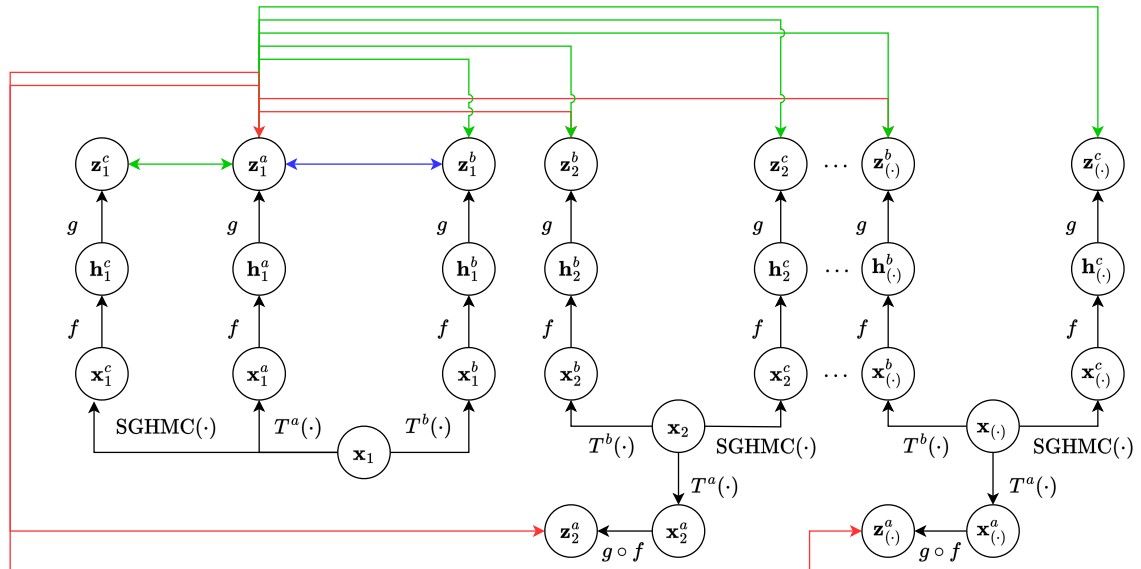

Figure 3: The schematic for the coupling of the proposed NRCC with the existing contrastive learning paradigms. Specifically, in this graphical workflow we focus on InfoNCE. However, the workflow can also be extended to BYOL with some modifications as highlighted in Figure 5. Given a set of $n$ samples in the current batch namely $\{\mathbf{x}_1, \mathbf{x}_2, \cdots, \mathbf{x}_n\}$, let us consider three transformations $T^a(\cdot), T^b(\cdot)$, and the SGHMC$(\cdot)$. The network is composed as $g \circ f$, where $f$ is the encoder and $g$ is the additional projection. We pass each of the three transformations of $\mathbf{x}_i$, respectively $\mathbf{x}_i^a$, $\mathbf{x}_i^b$, and $\mathbf{x}_i^c$, for $i = 1, 2, \cdots, n$, through $f$ and $g$ to obtain the corresponding $\mathbf{h}_i^{(\cdot)}$s and $\mathbf{z}_i^{(\cdot)}$s. With this setup, considering an exemplary case of $\mathbf{x}_1$, the positive pair driven component in traditional InfoNCE loss is shown by the Blue line between $\mathbf{z}_1^a$ and $\mathbf{z}_1^b$. The negative-pairs in the canonical InfoNCE are defined by Red lines between $\mathbf{z}_1^a$ and $\{\mathbf{z}_{(\cdot)}^b\} \setminus \{\mathbf{z}_{(1)}^b\}$. Finally, the additional NRCC regularizer component takes care of everything else as shown by the Green line. Specifically, it considers the pairs of $\mathbf{z}_1^a$ and $\{\mathbf{z}_{(\cdot)}^c\} \cup \{\mathbf{z}_{(\cdot)}^b\}$s.

## 3.6 Contrastive Clustering with NRCC Framework

### 3.6.1 Coupling NRCC with InfoNCE and BYOL

In the case of InfoNCE+NRCC, the implementation is fairly straightforward, given that the CL paradigm already employs negative pairs. We only pass an additional negative pair formed by SGHMC to find the regularized loss in (5) and then follow Chen et al. (2020b) for optimizing InfoNCE+NRCC. We demonstrate

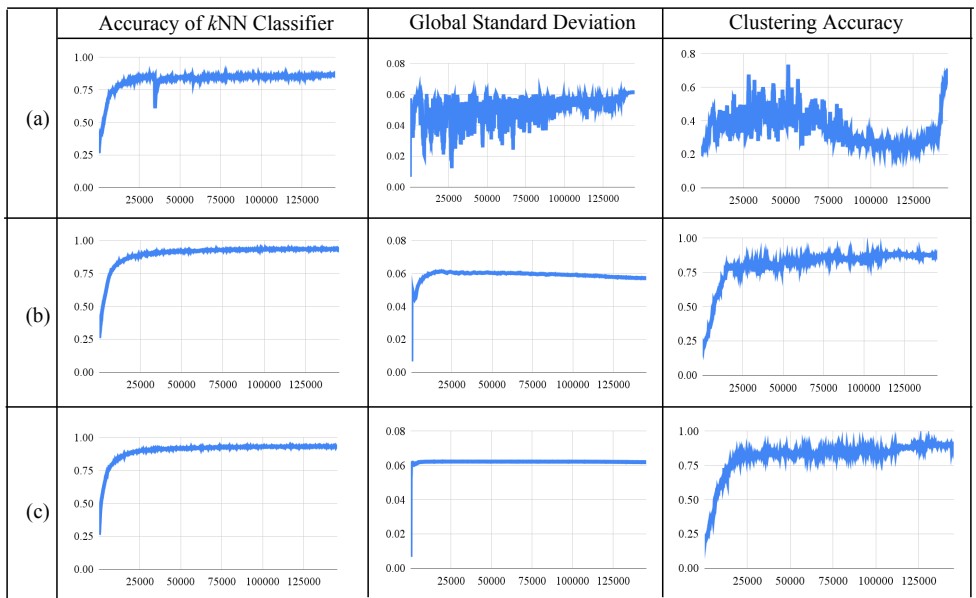

Figure 4: Plots of $k$-Nearest Neighbor ($k$NN) classification accuracy, Global Standard Deviation, and Clustering Accuracy against training iterations on ImageNet-10, for BYOL variants in terms of different couplings of negative pairs Specifically, the SGHMC augmented view $\mathbf{x}_j^c$ is passed through (a) an online network with prediction, (b) a target network with prediction, and (c) a typical target network without prediction. The third case (c) provides the best and most stable performance among the three alternatives.

this schematically in the following Figure 3. However, BYOL+NRCC brings additional challenges, given that no negative pair is involved in the original non-regularized formulation. If we consider the case of calculating $l_{i,a}^B$ then given $\mathbf{x}_j \neq \mathbf{x}_i$ to pass its SGHMC augmented view $\mathbf{x}_j^c$ one needs to consider two possible scenarios. First, $\mathbf{x}_j^c$ can be passed through either the online or the target network path. Second, if $\mathbf{x}_j^c$ follows the target branch, then it can either employ or discard an additional predictor at the end. Thus, to find a suitable network structure, we employ an ablation study following Chen & He (2021). We mainly consider three configurations. First, pass $\mathbf{x}_j^c$ to the online network. Second, feed $\mathbf{x}_j^c$ to the target network while using an additional prediction block. Third, pass $\mathbf{x}_j^c$ to the target network without a prediction block. For each of the three cases, over the training iteration on the ImageNet-10 dataset, we track the $k$-Nearest Neighbor ($k$NN) accuracy, global standard deviation, and clustering accuracy using GridShift (Kumar et al., 2022) following a UMAP dimensionality reduction, setting the number of neighbors $k$ as 5. We illustrate the findings in Figure 4 from which we can make three observations. First, if $\mathbf{x}_j^c$ passes through the online network, then the training is unstable as the momentum-based running average in the target network of BYOL offers the desired stability. Second, if $\mathbf{x}_j^c$ is passed through the target with a prediction block, then the global standard deviation slowly drops, indicating a collapse. Ideally, for a 256-variate projected Normal distribution, the global standard deviation should plateau at $1/\sqrt{256} \approx 0.06$ (Chen & He, 2021). This may be due to the fact that additional prediction in the target brings discrepancy in the learning, such that it is forcing cluster separability in the space spanned by $\{q(\mathbf{z}); \mathbf{z} \in \mathcal{Z}\}$ but not in the intended embedding space $\mathcal{H}$. Third, if $\mathbf{x}_j^c$ passes through a typical target network armed with stop gradient without additional prediction block, then all three indicators maintain stability and demonstrate improvement over training.

Now that we have the necessary clarity of effectively coupling BYOL and NRCC, let us follow up on the discussion in Section 3.2. If we consider the stop-gradient and predictor block in BYOL, then the actual gradients modify from their simplified versions in (12) to the following:

$$\frac{\partial l_-^B}{\partial \mathbf{z}_i^a} = \frac{\mathbf{z}_i^b}{2n}\left[\frac{\partial \bar{\mathbf{z}}_i^a}{\partial \mathbf{z}_i^a}\right] \text{ and } \frac{\partial l_+^B}{\partial \mathbf{z}_i^a} = 0. \tag{19}$$

**Algorithm 2:** The proposed InfoNCE+NRCC.

---

**Input:** Input dataset $\{\mathbf{x}_1, \mathbf{x}_2, \cdots, \mathbf{x}_N\}$, batch size $n$, learning rate $\beta$, weight of NRCC regularizer in the total loss $\lambda$, temperature parameter $\tau$, number of maximum iterations $\zeta$, and pool of transformations $\mathcal{T}$.

**Output:** The learned encoder $f$.

---

**1** Randomly initialize the parameters for $f$ and $g$.

**2** **while** *Maximum iteration $\zeta$ is not reached* **do**

**3**      Sample a random batch $\mathcal{N}$ of $n$ instances from the dataset.

**4**      **for** *all $i \in \{1, 2, \cdots, n\}$* **do**

**5**          Sample transformations $T^a, T^b \in \mathcal{T}$ and create $\mathbf{x}_i^a = T^a(\mathbf{x}_i)$ and $\mathbf{x}_i^b = T^b(\mathbf{x}_i)$.

**6**      **end**

**7**      Create positive pairs as $(\mathbf{x}_i^a, \mathbf{x}_i^b)$, for all $i \in \{1, 2, \cdots, n\}$.

**8**      Create negative pairs as $(\mathbf{x}_i^a, \mathbf{x}_j^k)$, for all $i, j \in \{1, 2, \cdots, n\}$, $\mathbf{x}_k^c \in \{\mathbf{x}_j^a, \mathbf{x}_j^b\}$ and $i \neq j$.

**9**      Find the SGHMC transformations $\mathbf{x}_i^c$ for all $i \in \{1, 2, \cdots, n\}$ using Algorithm 1.

**10**      Create NRCC negative pairs as $(\mathbf{x}_i^a, \mathbf{x}_j^c)$ where $i, j \in \{1, 2, \cdots, n\}$ and $i \neq j$.

**11**      Calculate $l^I$ using (5).

**12**      Update $f$ and $g$ using the gradients of $l^I$.

**13** **end**

**14** Discard $g$ and return $f$ as the feature encoder.

---

NRCC achieves its true potential when it is applied on the $\mathcal{Z}$-space of the target network in the BYOL architecture. In other words, while regularizing, we only need to account for the stop gradient and ignore the prediction block. This only impacts the negative gradient-induced attraction while the positive gradient-based repulsion remains the same as in (14). After revising (13), the regularized negative gradient takes the form:

$$\frac{\partial l_-^B}{\partial \mathbf{z}_i^a} = \frac{\mathbf{z}_i^b}{n} \left[ \frac{1}{2} \frac{\partial \bar{\mathbf{z}}_i^a}{\partial \mathbf{z}_i^a} + \lambda \frac{\exp(\langle \mathbf{z}_i^a, \mathbf{z}_i^b \rangle / \tau)}{\sum_{l=1}^n \exp(\langle \mathbf{z}_i^a, \mathbf{z}_l^b \rangle / \tau)} \right]. \tag{20}$$

The simplified formulations (12) and (13), when compared with their respective counterparts in (19) and (20) leads to two key observations. First, the canonical and regularized positive gradients remain exactly the same after considering the BYOL-specific heuristics. Second, as already mentioned in Section 3.2, the negative gradient of the canonical BYOL now has a multiplicative factor $\frac{\partial \bar{\mathbf{z}}_i^a}{\partial \mathbf{z}_i^a}$ that does not impact the additional force exerted by the regularizer. Thus, the motivation in favor of NRCC that we had mentioned earlier in Section 3.2 remains unaltered even if we account for the implementation-specific heuristics. We present the schematic workflow of BYOL+NRCC in Figure 5. We also describe InfoNCE+NRCC and BYOL+NRCC, respectively, in the following Algorithms 2 and 3. The code base is available at `https://github.com/abhisheka456/NRCC`.

### 3.6.2 Clustering in the NRCC Regularized CL Embedding Space

After optimizing the InfoNCE+NRCC or BYOL+NRCC loss, we can safely discard everything except the $f$, i.e., the original encoder. In the case of BYOL+NRCC, we only retain the $f$ from the target branch. Now, at the end of training, we get the cluster-friendly embedding. If we know the number of clusters beforehand, then we can apply a parametric technique like $k$-means to the learned $f$ space. Otherwise, we can project the embeddings to a lower-dimensional space through methods like UMAP or t-SNE (Van der Maaten & Hinton, 2008) that retain the local neighborhoods. Now, NRCC, along with SGHMC-based augmentation, manages to properly identify and separate the clusters in the embedding space. Thus, the revealed cluster structures are not distorted after additional neighborhood-preserving dimension reduction. This further enables the use of mode-seeking clustering methods like GridShift that do not require knowledge of the number of clusters.

**Algorithm 3:** The proposed BYOL+NRCC.

---

**Input:** Input dataset $\{\mathbf{x}_1, \mathbf{x}_2, \cdots, \mathbf{x}_N\}$, batch size $n$, learning rate $\beta$, weight of NRCC regularizer in the total loss $\lambda$, temperature parameter $\tau$, number of maximum iterations $\zeta$, and pool of transformations $\mathcal{T}$.

**Output:** The learned encoder $f$.

---

**1** Randomly initialize the parameters for online network $f$, $g$, and $q$ and target network $\bar{f}$ and $\bar{g}$.

**2 while** *Maximum iteration $\zeta$ is not reached* **do**

**3**     Sample a random batch $\mathcal{N}$ of $n$ instances from the dataset.

**4**     **for** *all $i \in \{1, 2, \cdots, n\}$* **do**

**5**         Sample transformations $T^a, T^b \in \mathcal{T}$ and create $\mathbf{x}_i^a = T^a(\mathbf{x}_i)$ and $\mathbf{x}_i^b = T^b(\mathbf{x}_i)$.

**6**     **end**

**7**     Create positive pairs as $(\mathbf{x}_i^a, \mathbf{x}_i^b)$, for all $i \in \{1, 2, \cdots, n\}$.

**8**     Find SGHMC transformations $\mathbf{x}_i^c$ for all $i \in \{1, 2, \cdots, n\}$ by Algorithm 1.

**9**     Create NRCC negative pairs as $(\mathbf{x}_i^a, \mathbf{x}_j^c)$ where $i, j \in \{1, 2, \cdots, n\}$ and $i \neq j$.

**10**     Calculate $l^B$ using (6).

**11**     Update online network $f$, $g$, and $q$ using the gradients of $l^B$.

**12**     Update target network $\bar{f}$ and $\bar{g}$ by momentum-based running average of online parameters.

**13 end**

**14** Discard everything except $\bar{f}$ and return that as the feature encoder.

---

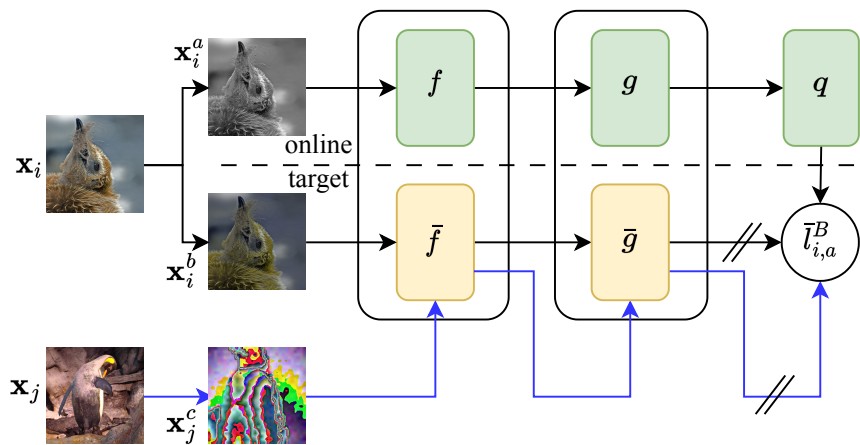

Figure 5: For BYOL+NRCC, the network architecture remains the same while the algorithm (original denoted in black) adds an extra pass (shown by the blue line) of the negative pair through the target network (the branch in green). A negative pair is created by taking an augmentation of $\mathbf{x}_i$ as $\mathbf{x}_i^a$ and the SGHMC transformation of another sample $\mathbf{x}_j$ as $\mathbf{x}_j^c$. To calculate the NRCC regularizer, the $\mathbf{x}_i^a$ is passed through the online network (the branch in green) to get $q(\mathbf{z}_i^a)$ while $\mathbf{x}_j^c$ goes through target to be mapped to $\bar{\mathbf{z}}_j^c$. The loss can then be calculated by (6) for updating online while ensuring stop gradient on target. The target is then updated using the momentum-driven running average of the online parameters as in Grill et al. (2020).

## 4 Experiments

### 4.1 Experimental Protocol

To ensure a fair comparison, we refrain from using pre-trained models in all our experiments. We use ResNet-34 as our backbone network following the conventional recommendation. Moreover, as per standard protocol (Kim & Ye, 2022; Chen et al., 2020b), we train the backbone for 1000 epochs with a batch size 256. The number of iterations of SGHMC is set to 1 as increasing it did not provide any considerable

Table 2: Ablation study on CIFAR-10 dataset in terms of NMI, ARI, ACC, highlighting how the regularization by NRCC, negative pair generation by SGHMC, data dimension reduction by UMAP, and clustering by GridShift, all when coupled together in the proposed framework, improve the performance for CL methods like InfoNCE and BYOL (best is boldfaced, second best underlined). Please note, DR: Dimension Reduction technique, CA: Clustering Algorithm, and Aug: traditional Augmentations.

| Comment | CL | NRCC | | DR | | CA | | Performance Indices | | |
|---|---|---|---|---|---|---|---|---|---|---|
| | | Aug | SGHMC | t-SNE | UMAP | $k$-means | GridShift | NMI | ACC | ARI |
| CL+$k$-means | InfoNCE | × | × | × | × | ✓ | × | 77.6 | 77.8 | 71.7 |
| | BYOL | × | × | × | × | ✓ | × | 81.7 | 89.4 | 79.0 |
| CL+t-SNE+GridShift | InfoNCE | × | × | ✓ | × | × | ✓ | 78.1 | 78.3 | 72.5 |
| | BYOL | × | × | ✓ | × | × | ✓ | 81.9 | 89.5 | 80.1 |
| CL+UMAP+GridShift | InfoNCE | × | × | × | ✓ | × | ✓ | 78.7 | 78.9 | 74.5 |
| | BYOL | × | × | × | ✓ | × | ✓ | 82.5 | 89.9 | 82.3 |
| CL+NRCC w/o SGHMC+$k$-means | InfoNCE | ✓ | × | × | × | ✓ | × | 80.4 | 85.1 | 80.2 |
| | BYOL | ✓ | × | × | × | ✓ | × | 84.5 | 90.6 | 84.1 |
| CL+NRCC w/o SGHMC+t-SNE+GridShift | InfoNCE | ✓ | × | ✓ | × | × | ✓ | 80.7 | 85.9 | 80.5 |
| | BYOL | ✓ | × | ✓ | × | × | ✓ | 84.9 | 91.1 | 84.8 |
| CL+NRCC w/o SGHMC+UMAP+GridShift | InfoNCE | ✓ | × | × | ✓ | × | ✓ | 81.5 | 86.7 | 82.4 |
| | BYOL | ✓ | × | × | ✓ | × | ✓ | 85.1 | 92.2 | 85.1 |
| CL+NRCC+$k$-means | InfoNCE | × | ✓ | × | × | ✓ | × | 86.8 | 92.3 | 85.2 |
| | BYOL | × | ✓ | × | × | ✓ | × | 87.6 | 92.8 | 87.0 |
| CL+NRCC+t-SNE+GridShift | InfoNCE | × | ✓ | ✓ | × | × | ✓ | 87.0 | 92.7 | 85.9 |
| | BYOL | × | ✓ | ✓ | × | × | ✓ | 88.1 | 93.3 | 87.2 |
| CL+NRCC+UMAP+GridShift (Ours) | InfoNCE | × | ✓ | × | ✓ | × | ✓ | 87.1 | 93.4 | 86.7 |
| | BYOL | × | ✓ | × | ✓ | × | ✓ | **88.4** | **93.8** | **87.9** |

performance boost. The rest of the hyper-parameters for the proposed method are tuned by grid search, and we detail the same in Appendix C. All the contending algorithms follow their originally recommended settings. For evaluating the clustering performance of the proposed methods, we consider four types of datasets, namely large-scale moderate resolution CIFAR-10, CIFAR-100 (Krizhevsky & Hinton, 2009), and STL-10 (Coates et al., 2011), moderate scale higher resolution subsets of ImageNet such as ImageNet-10 and ImageNet-Dogs (Russakovsky et al., 2015), large-scale higher resolution TinyImageNet (Le & Yang, 2015), and large scale moderate resolution long-tailed CIFAR-10-LT (Tang et al., 2020) and CIFAR-20-LT (Tang et al., 2020). Further details on datasets are provided in Appendix C. For CIFAR variants, we follow a similar architectural modification to ResNet-34 as suggested by SimCLR (Chen et al., 2020b). All contenders map the input to 256-dimensional embedding space while data reduction, if used, further decreases it to three such that mode-seeking algorithms can be applied. The code base for the proposed techniques can be found at `https://github.com/abhisheka456/NRCC`.

## 4.2 Ablation Study

We begin with an ablation study on the CIFAR-10 dataset to understand how the different components of the proposed framework impact its clustering performance. We break down the ablation study into four stages. First, setting the baseline performances of InfoNCE and BYOL when a $k$-means clustering is used. Second, evaluating the improvement of baseline clustering performances if t-SNE (Van der Maaten & Hinton, 2008) or UMAP is used for dimension reduction, followed by GridShift. Third, validate the significance of NRCC for obtaining a cluster-friendly embedding space by applying the regularizer with standard augmentation techniques. Fourth, gathering evidence in support of SGHMC as a fitting augmentation technique in the NRCC framework. In Table 2, we detail our findings in terms of three clustering performance evaluation metrics, namely Normalized Mutual Information (NMI) (Strehl & Ghosh, 2002), Clustering Accuracy (ACC) (Xie et al., 2016), and Adjusted Rand Index (ARI) (Hubert & Arabie, 1985). From Table 2, we can observe that the baseline performances in terms of all three indices improve when GridShift is applied on the lower dimensional space instead of $k$-means on the CL-embedding space. Moreover, the improvements by GridShift are more pronounced when UMAP is used for dimension reduction. We can make two critical conclusions from these findings. First, the mode-seeking algorithm on a lower-dimensional space not only removes the demand

for the number of clusters $k$ but also provides a performance boost. Second, if performed by conserving local neighborhoods, data dimensionality reduction may further improve representation quality by decreasing the redundancy of noisy features still present in the embedding space. The rest of the ablation study provides evidence in favor of NRCC, i.e., the proposed regularizer, by demonstrating how it, in general, improves the clustering performance of baseline algorithms. However, the true potential of NRCC is only realized when SGHMC is used for negative pair generation instead of traditional augmentation. This reaffirms SGHMC as an integral part of NRCC and highlights the necessary characteristics of a transformation that best suits a CL task. Moreover, UMAP is consistently found to be a better choice than t-SNE for translating the cluster-friendly embedding space onto a lower-dimensional one, i.e., suitable for effectively employing GridShift. Following these empirical observations, we safely fix the choice for our dimensionality reduction method to UMP and the clustering technique to GridShift. Thus, our prescription for an optimal DC technique takes the shape of Baseline+NRCC+GridShift.

Table 3: Comparison of clustering performance of the proposed NRCC+GridShift, coupled with InfoNCE and BYOL, against 20 other State-Of-The-Art (SOTA) DC methods in terms of ACC on six benchmark image datasets of varying scale and resolution. The best is boldfaced, the second best underlined, and the Margin is defined as the difference between the best proposed and its nearest existing SOTA.

| Method | CIFAR-10 | CIFAR-100 | STL-10 | ImageNet-10 | ImageNet-Dogs | TinyImageNet |
|---|---|---|---|---|---|---|
| IIC (Ji et al., 2019) | 61.7 | 25.7 | 49.9 | - | - | - |
| DCCM (Wu et al., 2019) | 62.3 | 32.7 | 48.2 | - | - | - |
| PICA (Huang et al., 2020) | 64.5 | 32.2 | - | - | - | - |
| CC (Li et al., 2021) | 79.0 | 42.9 | 85.0 | 89.3 | 42.9 | 14.0 |
| MiCE (Tsai et al., 2021) | 83.5 | 44.0 | 75.2 | - | 43.9 | - |
| GCC (Zhong et al., 2021) | 85.6 | 47.2 | 78.8 | 90.1 | 52.6 | 13.8 |
| TCL (Li et al., 2022) | 88.7 | 53.1 | 86.8 | 89.5 | 64.4 | - |
| TCC (Li et al., 2022) | 90.6 | 49.1 | 81.4 | 89.7 | 59.5 | - |
| C3 (Sadeghi et al., 2022) | 83.8 | 45.1 | - | 94.2 | 43.4 | 14.1 |
| SimCLR (Chen et al., 2020b) | 77.8 | 42.6 | 71.6 | 90.6 | 34.6 | 14.8 |
| MoCo (He et al., 2020) | 77.6 | 39.7 | 72.8 | - | 33.8 | 16.0 |
| SimSiam (Chen & He, 2021) | 85.6 | - | 71.6 | 92.1 | 67.4 | 20.3 |
| BYOL (Grill et al., 2020) | 89.4 | - | 82.5 | 93.9 | 69.4 | 19.9 |
| EBCLR (Kim & Ye, 2022) | 79.3 | 42.9 | 72.3 | 81.2 | 31.9 | 13.2 |
| PCL (Li et al., 2020a) | 87.4 | - | 41.0 | 90.7 | 41.2 | 15.9 |
| IDFD (Tao et al., 2021) | 81.5 | 42.5 | 75.6 | 95.4 | 59.1 | - |
| SCAN (Dang et al., 2021) | 88.3 | 44.0 | 80.9 | - | - | - |
| NMM (Dang et al., 2021) | 84.3 | 47.7 | 80.8 | - | - | - |
| CCES (Yin et al., 2023) | 81.2 | 44.2 | 84.7 | - | - | 19.6 |
| SACC (Deng et al., 2023) | 85,1 | 44.3 | 75.9 | 90.5 | 43.7 | - |
| InfoNCE+NRCC+GridShift (Ours) | 93.4 | 49.8 | 88.2 | 92.1 | 69.6 | 28.5 |
| BYOL+NRCC+GridShift (Ours) | **93.8** | **63.9** | **88.2** | **95.9** | **72.6** | **31.4** |
| Margin | **+3.2** | **+10.8** | **+1.8** | **+0.5** | **+3.2** | **+11.1** |

## 4.3 Performance Comparison on Benchmark Datasets

For the comparative study of clustering performance, we take six benchmark datasets, namely, CIFAR-10, CIFAR-20, STL-10, and the three ImageNet subsets. As competing algorithms, we consider 20 state-of-the-art (SOTA) methods spread across six different DC strategies. First, the non-CL-techniques, such as, IIC (Ji et al., 2019), DCCM (Wu et al., 2019) and PICA (Van Gansbeke et al., 2020). Second, the methods that directly produce cluster assignments, like CC (Li et al., 2021), MiCE (Tsai et al., 2021), GCC (Zhong et al., 2021), TCL (Li et al., 2022), TCC (Shen et al., 2021), and C3 (Sadeghi et al., 2022). Third, CL-methods that are not tailored for clustering but produce a generalized embedding that can be efficiently clustered, for example, SimClr (Chen et al., 2020b), EBCLR (Kim & Ye, 2022), MoCo (He et al., 2020), SimSiam (Chen & He, 2021), and BYOL (Grill et al., 2020). Fourth, CL methods tailored for clustering tasks, such as IDFD (Tao et al., 2021) and PCL (Li et al., 2020a). Fifth, multi-stage methods where a pre-training is followed by a clustering task-specific fine-tuning, like SCAN (Van Gansbeke et al., 2020) and NMM (Dang et al., 2021). Sixth, a couple of recent contrastive clustering methods explore the importance of better augmentation strategies in the context, namely, CCES (Yin et al., 2023) and SACC (Deng et al., 2023).

Table 4: Comparison of clustering performance of the proposed NRCC+GridShift, coupled with InfoNCE and BYOL, against 20 other State-Of-The-Art (SOTA) DC methods in terms of NMI on six benchmark image datasets of varying scale and resolution. The best is boldfaced, the second best underlined, and the Margin is defined as the difference between the best proposed and its nearest existing SOTA.

| Method | CIFAR-10 | CIFAR-100 | STL-10 | ImageNet-10 | ImageNet-Dogs | TinyImageNet |
|---|---|---|---|---|---|---|
| IIC (Ji et al., 2019) | 51.3 | - | 43.1 | - | - | - |
| DCCM (Wu et al., 2019) | 49.6 | 28.5 | 37.6 | - | - | - |
| PICA (Huang et al., 2020) | 56.1 | 29.6 | - | - | - | - |
| CC (Li et al., 2021) | 70.5 | 43.1 | 76.4 | 85.9 | 44.5 | 34.0 |
| MiCE (Tsai et al., 2021) | 73.7 | 43.6 | 63.5 | - | 42.3 | - |
| GCC (Zhong et al., 2021) | 76.4 | 47.2 | 68.4 | 84.2 | 49.0 | 34.7 |
| TCL (Li et al., 2022) | 81.9 | 52.9 | 79.9 | 87.5 | 62.3 | - |
| TCC (Li et al., 2022) | 79.0 | 47.9 | 73.2 | 84.8 | 55.4 | - |
| C3 (Sadeghi et al., 2022) | 74.8 | 43.4 | - | 90.5 | 44.8 | 33.5 |
| SimCLR (Chen et al., 2020b) | 77.6 | 40.9 | 62.4 | 80.6 | 35.9 | 33.8 |
| EBCLR (Kim & Ye, 2022) | 78.1 | 41.3 | 64.6 | 78.4 | 32.4 | 29.5 |
| MoCo (He et al., 2020) | 66.9 | 39.0 | 61.5 | - | 34.7 | 34.2 |
| SimSiam (Chen & He, 2021) | 78.6 | - | 65.9 | 83.1 | 58.3 | 35.1 |
| BYOL (Grill et al., 2020) | 81.7 | - | 71.3 | 86.6 | 63.5 | 36.5 |
| PCL (Li et al., 2020a) | 80.2 | - | 71.8 | 84.1 | 44.0 | 35.0 |
| IDFD (Tao et al., 2021) | 71.1 | 42.6 | 64.3 | 89.8 | 54.6 | - |
| SCAN (Dang et al., 2021) | 79.7 | 44.9 | 69.8 | - | - | - |
| NMM (Dang et al., 2021) | 74.8 | 48.4 | 69.4 | - | - | - |
| CCES (Yin et al., 2023) | 72.4 | 43.6 | 77.5 | - | - | 38.2 |
| SACC (Deng et al., 2023) | 76.5 | 44.8 | 69.1 | 87.7 | 45.5 | - |
| InfoNCE+NRCC+GridShift (Ours) | 87.1 | 61.5 | 80.5 | 84.3 | 64.2 | 43.5 |
| BYOL+NRCC+GridShift (Ours) | **88.4** | **65.8** | **80.7** | **90.6** | **68.4** | **47.8** |
| Margin | **+6.5** | **+12.9** | **+0.8** | **+0.1** | **+4.9** | **+9.6** |

Table 5: Comparison of clustering performance of the proposed NRCC+GridShift, coupled with InfoNCE and BYOL, against 20 other State-Of-The-Art (SOTA) DC methods in terms of ARI on six benchmark image datasets of varying scale and resolution. The best is boldfaced, the second best underlined, and the Margin is defined as the difference between the best proposed and its nearest existing SOTA.

| Method | CIFAR-10 | CIFAR-100 | STL-10 | ImageNet-10 | ImageNet-Dogs | TinyImageNet |
|---|---|---|---|---|---|---|
| IIC (Ji et al., 2019) | 41.1 | - | 29.5 | - | - | - |
| DCCM (Wu et al., 2019) | 40.8 | 17.3 | 26.2 | - | - | - |
| PICA (Huang et al., 2020) | 46.7 | 15.9 | - | - | - | - |
| CC (Li et al., 2021) | 63.7 | 26.6 | 72.6 | 82.2 | 27.4 | 7.1 |
| MiCE (Tsai et al., 2021) | 69.8 | 28.0 | 57.5 | - | 28.6 | - |
| GCC (Zhong et al., 2021) | 72.8 | 30.5 | 63.1 | 82.2 | 36.2 | 7.5 |
| TCL (Li et al., 2022) | 78.0 | 35.7 | 75.7 | 83.7 | 51.6 | - |
| TCC (Li et al., 2022) | 73.3 | 31.2 | 68.9 | 82.5 | 41.7 | - |
| C3 (Sadeghi et al., 2022) | 70.7 | 27.5 | - | 86.1 | 28.0 | 6.5 |
| SimCLR (Chen et al., 2020b) | 71.7 | 25.8 | 51.6 | 82.1 | 23.8 | 6.7 |
| EBCLR (Kim & Ye, 2022) | 72,4 | 25.9 | 53.8 | 77.6 | 19.3 | 5.8 |
| MoCo (He et al., 2020) | 60.8 | 24.2 | 52.4 | - | 19.7 | 8.0 |
| SimSiam (Chen & He, 2021) | 73.6 | - | 57.2 | 83.3 | 50.1 | 9.4 |
| BYOL (Grill et al., 2020) | 79.0 | - | 65.7 | 87.2 | 54.8 | 10.0 |
| PCL (Li et al., 2020a) | 76.6 | - | 67.0 | 82.2 | 29.9 | 8.7 |
| IDFD (Tao et al., 2021) | 66.3 | 26.4 | 57.5 | 90.1 | 41.3 | - |
| SCAN (Dang et al., 2021) | 77.2 | 28.3 | 64.6 | - | - | - |
| NMM (Dang et al., 2021) | 70.9 | 31.6 | 65.0 | - | - | - |
| CCES (Yin et al., 2023) | 69.4 | 30.1 | 73.1 | - | - | 12.5 |
| SACC (Deng et al., 2023) | 72.4 | 28.2 | 62.6 | 84.3 | 28.5 | - |
| InfoNCE+NRCC+GridShift (Ours) | 86.7 | 34.5 | 76.0 | 83.4 | 54.1 | 15.7 |
| BYOL+NRCC+GridShift (Ours) | **87.9** | **49.2** | **76.1** | **90.2** | **59.4** | **17.6** |
| Margin | **+8.9** | **+13.5** | **+0.4** | **+0.1** | **+4.6** | **+5.1** |

We take three performance indices, namely ACC, NMI, and ARI, to measure the clustering quality of an algorithm. In Table 3, we report the performance in terms of ACC while Tables 4 and 5 respectively tabulates

Table 6: Comparison of clustering performance of the proposed NRCC+GridShift, coupled with InfoNCE and BYOL, against MoCo and baseline BYOL, in terms of NMI, ACC, and ARI, on two long-tailed image datasets. The best is boldfaced and Margin is defined as the difference between the best proposed and its nearest existing SOTA.

| Method | CIFAR-10-LT | | | CIFAR-20-LT | | |
|---|---|---|---|---|---|---|
| | NMI | ACC | ARI | NMI | ACC | ARI |
| MoCo | 46.7 | 33.4 | 27.7 | 31.2 | 28.2 | 16.1 |
| BYOL | 51.6 | 41.3 | 30.8 | 41.9 | 34.6 | 22.3 |
| BYOL+NRCC+GridShift (Ours) | **55.6** | **44.2** | **35.3** | **45.3** | **38.6** | **28.3** |
| Margin | **+5.0** | **+2.9** | **+4.5** | **+3.4** | **+4.0** | **+5.0** |

NMI and ARI measures. Closer inspection of Table 3 reveals that the proposed BYOL+NRCC+GridShift obtains the best performance on all the six datasets, improving accuracy by 5.10% on average from the nearest contenders among the 20 SOTA methods. The proposed InfoNCE+NRCC+GridShift also attains the second position on four datasets. This is an expected outcome given that BYOL is known to be a superior representation learner (Grill et al., 2020). Similar behavior is reflected in Tables 4 and 5 as well. Specifically, BYOL+NRCC+GridShift maintains its best position in terms of both NMI and ARI over all six datasets, leading ahead from its nearest contender, respectively, by 5.80% and 5.43% on average. The other NRCC variant, InfoNCE+NRCC+GridShift, also performs consistently, standing second over five datasets in terms of NMI and four datasets when ARI is considered. These findings unanimously testify that NRCC indeed provides a cluster-friendly embedding space that, when preserved by data dimensionality reduction through local neighborhood preserving UMAP, can significantly improve performance by GridShift without requiring the knowledge of the number of clusters. The generally improved representations learned by BYOL over InfoNCE are also further validated through their clustering performance.

### 4.4 Performance Comparison on Long-tailed Datasets

Up to this point, we have only validated the performance of the proposed BYOL+NRCC+GridShift on datasets with uniform-sized clusters. However, in reality, it is common to find datasets with imbalanced clusters or long-tailed cluster distributions. Thus, we consider two imbalanced datasets, namely, CIFAR-10-LT and CIFAR-20-LT, to validate NRCC's effectiveness. In Table 6, we compare the clustering performance of the BYOL+NRCC+GridShift against MoCo and baseline BYOL in terms of NMI, ACC, and ARI. From Table 6, we can observe that BYOL+NRCC+GridShift retains its best performance, followed by BYOL, while MoCo performs the worst among the three. Specifically, applying NRCC+GridShift improves NMI, ACC, and ARI, respectively, by 4.2%, 3.45%, and 4.75%, on average, over the baseline algorithm. Thus, as per the empirical evidence, we can safely conclude that even in the case of imbalanced clusters, NRCC is still capable of finding an embedding space where independent cluster structures are accurately preserved.

Table 7: Time taken in hours to train BYOL and BYOL+NRCC on the same device with four V100 GPUs.

| Dataset | BYOL | BYOL+NRCC (Ours) | Increment in Time | Increment in ACC |
|---|---|---|---|---|
| CIFAR-10 | 5h 33m | 8h 17m | 1.49x | 4.9% |
| STL-10 | 2h 37m | 3h 42m | 1.41x | 6.9% |
| ImageNet-10 | 5h 12m | 7h 18m | 1.40x | 2.1% |
| ImageNet-Dogs | 3h 17m | 4h 48m | 1.46x | 4.6% |
| TinyImageNet | 37h 11m | 52h 17m | 1.40x | 57.8% |
| Average Increment in Time for NRCC | | | | **1.43x** |
| Average Increment in ACC for NRCC | | | | **7.5%** |

### 4.5 Computational Overhead of the Proposed Clustering Methods

We have established the improved performance of BYOL+NRCC+GridShift, but to be sure of its usefulness, we need to study its computational feasibility compared to the baseline BYOL. Specifically, we want to know

if the probable increment in the proposed technique's training cost is negligible compared to the obtained gain in performance. We use the same computing setup with four V100 GPUs while calculating the time in hours to ensure fairness. Specifically, in Table 7, we compare the training time of baseline BYOL against the proposed BYOL+NRCC on five benchmark datasets, namely, CIFA-10, STL-10, and three ImageNet variants. We use the same computing setup with four V100 GPUs while calculating the time in hours to ensure fairness. Table 7 suggests that NRCC only puts 1.45x times computational overhead on the baseline on average, largely outmatched by the commendable performance improvement of 7.5% in clustering accuracy.

We undertake a follow-up study with similar computational settings to understand the potential speed and performance benefits of using SGHMC in NRCC. Specifically, we investigate the applicability of the two predecessors of SGHMC, namely Hamiltonian Monte Carlo (HMC) and the regular Monte Carlo (MC) sampling techniques. In Table 8, we summarize our findings regarding the training time and Accuracy of the three BYOL+NRCC variants when trained on CIFAR-10. We recall from Section 4.1 that in our implementation, SGHMC has only been iterated once. Hence, in our first experiment, we use a single iteration of HMC and MC to compare their performance against SGHMC. From Table 8 we can observe that in both cases SGHMC offers a better performance at a slightly higher computational cost. Regular MC though fastest among the three performs the worst by lagging 5.01% in Accuracy compared to SGHMC. On the other hand, though HMC and SGHMC share similar comparable costs, SGHMC outperforms HMC by 2.66%. Subsequently, to establish a fair setting, we increase the number of iterations for MC and HMC such that their performances closely match that of SGHMC. In this case, the performance gap between the three sampling strategies decreases while SGHMC becomes the most economical option by a large margin. Specifically, both HMC and MC reach a lower Accuracy compared to SGHMC while accruing increased costs of $1.89\times$ and $1.37\times$, respectively. In conclusion, SGHMC offers a more effective trade-off between performance and computational cost compared to its predecessors, when used in NRCC.

Table 8: Time taken in hours to train BYOL and BYOL+NRCC on the same device with four V100 GPUs on the CIFAR-10 dataset when NRCC uses SGHMC, Hamiltonian Monte Carlo (HMC), and regular Monte Carlo (MC) for generating negative pairs. Accuracy is abbreviated as ACC.

| Method | Comment | Time | ACC | Against Baseline | |
| --- | --- | --- | --- | --- | --- |
| | | | | Time | ACC |
| BYOL (Grill et al., 2020) | For reference | 5h 33m | 89.4 | - | - |
| BYOL+NRCC (single iteration SGHMC) | Proposed method, here Baseline | 8h 17m | **93.8** | - | - |
| BYOL+NRCC (single iteration HMC) | - | 7h 36m | 91.3 | +41m ($0.91\times$) | -2.66% |
| BYOL+NRCC (single iteration MC) | - | 5h 47m | 89.1 | +2h 30m ($0.69\times$) | -5.01% |
| BYOL+NRCC (multiple iterations HMC) | Closest to baseline ACC[*] | 15h 44m | 93.6 | -7h 27m ($1.89\times$) | -0.21% |
| BYOL+NRCC (multiple iteration MC) | Closest to baseline ACC[*] | 11h 24m | 93.2 | -3h 7m ($1.37\times$) | -0.63% |
| [*]: Further increment in iterations did not improve ACC. | | | | | |
| +: Better than baseline. -: Worse than Baseline. | | | | | |

## 4.6 Coupling NRCC with Other State-of-the-arts CL Methods

Up to this point, we restrict our discussion to InfoNCE and BYOL, two distinct foundational CL methods. Over the past few years, some other CL methods such as SimSiam Chen & He (2021) and Barlow Twins Zbontar et al. (2021) also gained popularity due to their superiority over the predecessors in some downstream tasks like classification and object detection. SimSiam can be thought of as a variant of BYOL that discards momentum-based updates. However, Chen & He (2021) agree that such a change may sacrifice accuracy. This may be caused by a slightly irregular feature embedding that does not preserve the cluster structures to the expected extent. On the other hand, Barlow Twins may compromise discriminative features in an attempt to reduce redundancy. This may negatively impact clustering performance which can worsen under incompatible choices of augmentation. Hence, coupling NRCC may not be proven beneficial for such baselines that are not canonically supportive of clustering performance. To validate this intuition, we consider an ablation study on the CIFAR-10 dataset in the following Table 9. We can see from Table 9 that SimSiam and Barlow Twins both perform better than SimCLR but worse than BYOL. Moreover, after coupling NRCC, both SimSiam and Barlow Twins perform worse than the two proposed techniques. Thus, NRCC aids both SimSiam and Barlow Twins in improving their clustering performance but fails to totally compensate for

their bias against such a task. In conclusion, NRCC improves the clustering performance of CL methods but the gain may be limited by the baseline algorithm.

Table 9: Coupling NRCC with other state-of-the-art CL methods.

| Method | ACC | ARI | NMI |
|---|---|---|---|
| SimSiam | 85.6 | 73.6 | 78.6 |
| Performance compared to SimCLR | +10% | +2.6% | +1.2% |
| Performance compared to BYOL | -4.2% | -6.8% | -3.7% |
| BarlowTwins | 87.3 | 74.2 | 78.4 |
| Performance compared to SimCLR | +12.2% | +3.4% | +1.0% |
| Performance compared to BYOL | -2.3% | -6.0% | -4.0% |
| SimSiam+NRCC+GridShift | 87.1 | 75.4 | 80.3 |
| Performance compared to InfoNCE+NRCC+GridShift (Ours) | -6.7% | -13.0% | -7.8% |
| Performance compared to BYOL+NRCC+GridShift (Ours) | -7.1% | -14.2% | -9.1% |
| BarlowTwins+NRCC+GridShift | 90.4 | 80.3 | 82.7 |
| Performance compared to InfoNCE+NRCC+GridShift (Ours) | -3.2% | -7.3% | -5.0% |
| Performance compared to BYOL+NRCC+GridShift (Ours) | -0.4% | -1.3% | -1.4% |

## 4.7 The Scalability of NRCC

To evaluate the scalability of the proposed NRCC we conduct a couple of experiments. The first experiment is focused on large-scale datasets with a high number of clusters such as the full ImageNet-1k. The second experiment investigates the applicability of NRCC with larger backbone networks such as the deeper variants of ResNet and Visual Transformers (ViT). In both cases, we consider the BYOL+NRCC+GridShift variant as our proposed method of choice given it maintains a consistent performance improvement over InfoNCE+NRCC+GridShift. The following Table 10 documents the findings of these two experiments.

In the first experiment with ImageNet-1k, we follow the experimental protocol described in Li et al. (2020a) and train a ResNet-50 for 200 epochs. To ensure a fair comparison, we only consider the state-of-the-art methods that have a reported result (either in the original paper or in a later reference such as Li et al. (2020a)) on ImageNet-1k in terms of Adjusted Mutual Information (AMI) under the common protocol (Li et al., 2020a). From the top half of Table 10 we can observe that even when the number of clusters is increased in ImageNet-1k the proposed BYOL+NRCC+GridShift sustains a commendable lead in Adjusted Mutual Information (AMI) over the state-of-the-art contenders. Specifically, the proposed BYOL+NRCC+GridShift method achieves a 5.9% improvement in AMI from its closest and most recent contender PIPCDR (Kumar & Lee, 2025). The experiment also shows that if we replace the traditional ResNet-50 backbone with a better performing ViT-Small network then a further 1.2% performance gain can be achieved for our method. Moreover, in comparison to the BYOL baseline, using ViT-Small the proposed NRCC regularizer can provide a 11.9% boost in AMI. Thus we can safely establish NRCC as scalable to a large number of clusters.

In the second experiment, we consider the ImageNet-Dogs dataset and vary the backbone network of BYOL+NRCC+GridShift between two ResNet and three ViT variants. Specifically, we take ResNet-34 and ResNet-50, the two widely popular residual backbones in contrastive learning literature, which are also used throughout this paper. We further consider the more recent ViT-Tiny, Vit-Small, and ViT-Base networks as backbones that gradually increase the number of parameters from 5.8M to 86M. For fairness, all five networks are trained under the same experimental protocol detailed in Section 4.1 and Appendix C. If we summarize the bottom part of Table 10 that presents the results of this second experiment then we can observe four interesting factors. First, changing the backbone from ResNet-34 to the larger ResNet-50 does not improve the performance but slightly decreases all three metrics. Second, Vit-Tiny suffers a somewhat limited performance deterioration of 2.7% (compared to the best performer Vit-Small) on average with only 26% of parameters. Third, Vit-Small and ResNet-34 though share an almost equal number of parameters the ViT has an improved performance in terms of all three indices, making it the best among the five choices. Fourth, similar to ResNet increasing the number of parameters from Vit-Small to ViT-Base also acts against the performance. We can make two conclusions from these observations. First, ViT owing to its generally better performance than ResNet has the potential to become a primary backbone for contrastive clustering. Second, at least for most clustering tasks, blindly increasing model capacity without accounting for the data availability and problem complexity may not be a straightforward strategy for finding success. This is evident

from the consistent performance of similar-sized models such as ViT-Small and ResNet-34 on moderate-scale problems like ImageNet subsets while the larger ResNet-50 finds success on more complex ImageNet-1k as seen in the previous experiment.

Table 10: The scalability of the proposed BYOL+NRCC+GridShift contrastive clustering method. The best result is boldfaced while the second best is underlined. Note that the abbreviation NoP stands for the Number of Parameters in the backbone network.

| The performance of BYOL+NRCC+GridShift on ImageNet-1k compared to the state-of-the-art. | | | | |
|---|---|---|---|---|
| Experiment | Dataset | Method | Backbone (NoP) | AMI |
| Large dataset | ImageNet-1k | DeepCluster (Li et al., 2020a)[1]
MoCo (Li et al., 2020a)[2]
PCL (Li et al., 2020a)
ProPos (Huang et al., 2022)
PIPCDR (Kumar & Lee, 2025)
BYOL+NRCC+GridShift (Ours) | ResNet-50 [25.6M] | 28.1
28.5
41.0
52.5
54.2
57.4 |
| | | BYOL
BYOL+NRCC+GridShift (Ours) | ViT-Small (22.2M) | 51.9
**58.1** |

[1] DeepCluster Original article Caron et al. (2018), [2] MoCo original article He et al. (2020).

| The performance of BYOL+NRCC+GridShift on ImageNet-Dogs when larger backbone networks are used. | | | | | | |
|---|---|---|---|---|---|---|
| Experiment | Dataset | Method | Backbone (NoP) | NMI | ACC | ARI |
| Large backbone | ImageNet-Dogs | BYOL+NRCC+GridShift (Ours) | ResNet-34 (22.7M)
ResNet-50 (25.6M)
ViT-Tiny (5.8M)
ViT-Small (22.2M)
ViT-Base (86M) | 68.4
67.6
66.9
**68.8**
68.5 | 72.6
72.2
72.0
**73.8**
72.6 | 59.4
59.3
58.3
**60.1**
59.5 |

## 5 Conclusion and Future Works

We recognize the uncontrolled interplay between positive pair-induced attraction and, if present, negative pair-influenced repulsion as critical factors leading to collapsed or fragmented clusters in CL-based DC methods. In response, we introduce the NRCC regularizer, which carefully balances these two counteracting forces while enhancing them to ensure that the inherent cluster structure in the dataset remains intact in the embedding space. To achieve this, we re-envision SGHMC, a sampling method initially designed to improve the computational efficiency of Hamiltonian Monte Carlo for online inference, as an approximately invariant augmentation strategy in CL. As per our knowledge, this is the first work that aims to repurpose a sampling method such as SGHMC, as a data augmentation strategy. To further justify the aptitude of SGHMC as an augmentation in the CL framework we theoretically establish its approximately invariant property and empirically demonstrate its ability of curating hard negative pairs following our criteria. Utilizing SGHMC-generated negative pairs in NRCC introduces a density-aware repulsive force, facilitating the learning of a cluster-friendly embedding. Consequently, the representation acquired by NRCC can be effectively projected to a lower-dimensional space using a local neighborhood-preserving method. This opens up opportunities to apply mode-seeking algorithms that do not necessitate prior knowledge of the number of clusters.

There are two limitations of the current work that open potential future avenues of research. First, unlike end-to-end methods (Huang et al., 2022), we take a modular approach, where three subsequent steps for embedding learning, dimensionality reduction, and mode-seeking clustering are used. On one hand, such an approach offers flexibility allowing the user to employ state-of-the-art neighborhood-preserving dimensionality reduction techniques and clustering algorithms of choice to further improve application-specific performance. Moreover, unlike existing approaches, this modularity enables our framework to allow mode-seeking clustering algorithms that remove the dependency on the number of clusters. On the other hand, if a perfect harmony among the different choices for the components is not met, then that may restrict the framework from performing at its true potential. To alleviate this issue, a balance can be achieved by simultaneously performing the dimensionality reduction task in the CL embedding learning framework. Specifically, instead of directly using an external UMAP-type technique, one may emulate the same through an additional auxiliary loss in the CL objective itself. Such an approach may further benefit from shifting the problem to the graph contrastive learning domain, where representing the neighborhood and the modes can become easier. However, this

may not be straightforward given the additional computational overhead, algorithmic complexity, modified definition of positive and negative pairs, and the enhanced difficulty of directly finding the cluster structures in a low-dimensional embedding space. Second, even though SGHMC stands out as an excellent choice for augmentation in CL methods for DC, it still poses a potential threat to the time complexity due to its iterative nature. Our comparative study in Table 7 suggests that NRCC only introduces a minute computational overhead of 1.43x times over BYOL while performing 7.5% better on average than the canonical method. However, considering there may still be room for improvements, a future research direction may investigate better alternatives to SGHMC that can produce approximately invariant augmentations for hard negative pairs in a more comparatively economical way.

## Acknowledgment

This research has been conducted with the financial support of the European Union under the RE-FRESH - Research Excellence For Region Sustainability and High-tech Industries project number CZ.10.03.01/00/22_003/0000048 via the Operational Program Just Transition.

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

## A    Notations

The following Table 11 describes the different mathematical notations used throughout the paper. Apart from these, for the proofs of Proposition 3.1 and Theorem 3.6 (in Appendix B.1 and B.2 respectively) any additionally required notations and definitions are clearly described topically and are not carried forward elsewhere. Thus, for simplicity we discard them from Table 11.

Table 11: Mathematical notations used in this article according to the order of appearance.

| Notation | Description |
|---|---|
| $N, \mathcal{X}$ | $N$ data samples residing in the native space $\mathcal{X}$. |
| $\mathbb{N}_+, \mathbb{R}$ | The set of all positive integers and the set of real numbers. |
| $d, \bar{d}$ | The dimensionality of the native space $\mathcal{X}$ and the embedding space. |
| $\mathcal{H}$ | The embedding space of dimensionality $\bar{d}$. |
| $f$ | An encoder mapping from $\mathcal{X}$ to $\mathcal{H}$. |
| $\mathbf{x}, \mathbf{h}$ | A sample in native space from the set $\mathcal{X}$ and the embedding space $\mathcal{H}$. |
| $\mathcal{T}, T^a, T^b$ | The set of all possible augmentations and two augmentations in $\mathcal{T}$ |
| $\mathbf{x}^a(\mathbf{x}^b)$ | The sample $\mathbf{x}$ transformed by $T^a(T^b)$. |
| $\mathcal{N}, n$ | A mini-batch $\mathcal{N}$ of size $n$. |
| $\mathcal{P}_+(\mathcal{P}_-)$ | The set of all positive (negative) pairs in a mini-batch. |
| $\mathcal{P}_-^a(\mathcal{P}_-^b)$ | The set of negative pairs w.r.t. $T^a(T^b)$. |
| $g$ | Additional dense layer to map the embedding space to another space (Chen et al., 2020b). |
| $\mathbf{z}, \mathcal{Z}$ | The mapped sample $g(\mathbf{h})$ and the space spanned by the mapping $g$. |
| $\tau, \text{sim}(\cdot, \cdot)$ | Temperature parameter controlling the extent of similarity $\text{sim}(\cdot, \cdot)$, e.g. inner product $\langle \cdot, \cdot \rangle$. |
| $l^I(l_{i,a}^I, l_{i,b}^I)$ | InfoNCE loss function (w.r.t. $i$-th sample and augmentation $T^a(T^b)$). |
| $q$ | Additional prediction block in target branch of BYOL (Grill et al., 2020). |
| $l^B(l_{i,a}^B, l_{i,b}^B)$ | BYOL loss function (w.r.t. $i$-th sample and augmentation $T^a(T^b)$). |
| $\overline{(\cdot)}$ | Mappings and outputs for target branch of BYOL (Grill et al., 2020). |
| $\bar{l}(\cdot)$ | NRCC regularized loss functions. |
| $\lambda$ | Hyper-parameter for the relative weight of the NRCC regularizer. |
| $\mathbf{z}^c, \bar{\mathbf{z}}^c$ | Arbitrary augmented view of $\mathbf{x} \in \mathcal{N}$. |
| $\mathbb{P}, X, \hat{\mathbb{P}}_N$ | The probability distribution on space $\mathcal{X}$, a random variable following $\mathbb{P}$, and empirical distribution based on $N$ i.i.d. observations of $X$s. |
| $\phi$ | A map $\mathcal{T} \times \mathcal{X} \to \mathcal{X}$. |
| $\mathbf{e}$ | An identity augmentation $\mathbf{e} \in \mathcal{T}$. |
| $\neq_d$ | Non-identically distributed. |
| $\text{Conv}_{w,b}^\sigma, C, C'$ | Convolution operation with weight $w$, bias $b$, and activation $\sigma$ mapping from $C$ channels to $C'$. |
| $\text{FC}_{W,b}^\sigma$ | Fully connected layer with activation $\sigma$, weight $W$, and bias $b$. |
| $P$ | A typical padding operation. |
| $M, L$ | There are $M$ residual block each of depth $L$ in the deep network under concern. |
| $\Psi_\Omega^\sigma, \Omega$ | The embedding function realized through a deep network with the parameter space $\Omega$. |
| $\text{Id}$ | Identity mapping. |
| $B^c, B^f, D, \pi, \gamma$ | Constants greater than zero, used for various bounds. Note that $D$ is controlled by two non-negative real sequences $\{a_n\}_{n \in \mathbb{N}}$ and $\{b_n\}_{n \in \mathbb{N}}$. |
| $\mathbb{Q}, Y, \hat{\mathbb{P}}_{T,N}$ | The probability distribution of $\mathcal{T}$, a random variable following $\mathbb{Q}$-augmented distribution, and empirical distribution based on $N$ i.i.d. copies of $Y$. |
| $\mathbb{P}_\Omega$ | Probability distribution over pairwise distance characterized by $\Omega$. |
| $\zeta, \delta_1, \delta_2, \delta_3$ | Control parameters of SGHMC. |
| $\mathbf{p}, \mathbf{s}$ | Intermediate vectors associated with SGHMC. |
| $\mathbb{P}_\Omega^n$ | An empirical expectation of $\mathbb{P}_\Omega$ over the batch. |
| $\mathcal{W}_1, \mathcal{W}_2$ | 1 and 2-Wasserstein distances. |

# B  Notes on Theoretical Analysis

In this Section, we provide the detailed proofs of Proposition 3.1 and Theorem 3.6.

## B.1  Proof of Proposition 3.1

Let us start by observing that,

$$\left\| \mathbb{E}_{X,T} \Psi_\Omega^\sigma(\hat{\mathbb{P}}_{T,N}) - \mathbb{E}_X \Psi_\Omega^\sigma(\mathbb{P}) \right\|_2 \leq \mathbb{E}_T \mathcal{W}_1(\Psi_\Omega^\sigma(\hat{\mathbb{P}}_{T,N}), \Psi_\Omega^\sigma(\mathbb{P})),$$

where $\mathcal{W}_1$ is the 1-Wasserstein distance, defined as $\mathcal{W}_1(\alpha,\beta) = \inf_{\gamma \in \Gamma(\alpha,\beta)} \int_{\mathcal{H} \times \mathcal{H}} ||\mathbf{x} - \mathbf{y}|| d\gamma(\mathbf{x}, \mathbf{y})$, given $\mathcal{H}$ as the underlying Polish embedding space. Also, $\Gamma$ is the class of measure couples whose marginals turn out to be $\alpha$ and $\beta$. The inequality is due to the Kantorovich-Rubinstein duality (Chen et al., 2020a).

Now, let $\mathcal{L}_{\mathcal{X},\mathcal{H}}^\pi$ denote the class of $\pi$-Hölder functions mapping from $\mathcal{X}$ to $\mathcal{H}$, where $\pi > 0$. Given $l \in \mathcal{L}_{\mathcal{X},\mathcal{H}}^\pi$ and measures $\alpha, \beta$ defined on the input space, using the triangle inequality we get,

$$|\mathcal{W}_1(\Psi_\Omega^\sigma(\alpha), \Psi_\Omega^\sigma(\beta)) - \mathcal{W}_1(l(\alpha), l(\beta))| \leq \mathcal{W}_1(\Psi_\Omega^\sigma(\alpha), l(\alpha)) + \mathcal{W}_1(l(\beta), \Psi_\Omega^\sigma(\beta)).$$

Also, $\mathcal{W}_1(\Psi_\Omega^\sigma(\alpha), l(\alpha)) \lesssim ||\Psi_\Omega^\sigma - l||_\infty$, where the suppressed constant is $\tilde{d}^{\frac{1}{2}}$. Moreover, since there exists a constant $\Lambda > 0$, such that $\mathcal{L}_{\mathcal{X},\mathcal{H}}^\pi$ is a subset of the class of $\Lambda$-Lipschitz functions, one can argue,

$$\mathcal{W}_1(l(\alpha), l(\beta)) = \inf_{\gamma \in \Gamma'(\alpha,\beta)} \int_{\mathcal{X} \times \mathcal{X}} \left\| l(\mathbf{x}) - l(\mathbf{y}) \right\| d\gamma(\mathbf{x}, \mathbf{y}) \leq \Lambda \inf_{\gamma \in \Gamma'(\alpha,\beta)} \int_{\mathcal{X} \times \mathcal{X}} \|\mathbf{x} - \mathbf{y}\| d\gamma(\mathbf{x}, \mathbf{y}) = \Lambda \, \mathcal{W}_1(\alpha, \beta).$$

We now combine the findings till this point to get $\mathcal{W}_1(\Psi_\Omega^\sigma(\alpha), \Psi_\Omega^\sigma(\beta)) - \Lambda \, \mathcal{W}_1(\alpha, \beta) \lesssim ||\Psi_\Omega^\sigma - l||_\infty$.

Given any $l \in \mathcal{L}_{\mathcal{X},\mathcal{H}}^\pi$, and the activation function being ReLU, there exists a $\Psi_\Omega^\sigma$ with $\mathcal{O}(M)$ residual blocks, each with $\mathcal{O}(1)$ channels and of depth $\mathcal{O}(\log M)$, such that $||\Psi_\Omega^\sigma - l||_\infty \lesssim M^{-\frac{\pi}{d}}$ (Oono & Suzuki, 2019). The suppressed constant in the inequality involves $\tilde{d}$, while the result holds for any filter size $\in \{2, \cdots, d\}$. Considering our case,

$$\mathcal{W}_1(\Psi_\Omega^\sigma(\hat{\mathbb{P}}_{T,N}), \Psi_\Omega^\sigma(\mathbb{P})) - \Lambda \, \mathcal{W}_1(\hat{\mathbb{P}}_{T,N}, \mathbb{P}) \lesssim M^{-\frac{\pi}{d}}.$$

If exact invariance holds, i.e. $TX =_d X$ then, $\hat{\mathbb{P}}_{T,N}$ acts as a consistent estimator of $\mathbb{P}$ under the Wasserstein distance. In other words, the error will plunge to zero as $N$ increases. However, the departure from such a scenario would hinder the shrinkage of the discrepancy even in a large sample regime. To demonstrate the same, we write, $\mathcal{W}_1(\hat{\mathbb{P}}_{T,N}, \mathbb{P}) \leq \mathcal{W}_1(\hat{\mathbb{P}}_{T,N}, \hat{\mathbb{P}}_N) + \mathcal{W}_1(\hat{\mathbb{P}}_N, \mathbb{P})$, where the term $\mathcal{W}_1(\hat{\mathbb{P}}_N, \mathbb{P}) \xrightarrow{a.s.} 0$ as $N \to \infty$ (Weed & Bach, 2019). Further, in a non-asymptotic sense, it is upper bounded by $\mathcal{O}(N^{-1/\max(d,2)})$, with high probability. Therefore, essentially the error committed during estimation is the disagreement between the two plug-in estimators. This will give us the extent of deviation due to augmentation. Following Sriperumbudur et al. (2012), we can also write,

$$\mathcal{W}_1(\hat{\mathbb{P}}_{T,N}, \hat{\mathbb{P}}_N) = \sup_{f \in \mathcal{L}_{\mathcal{X}}^1} \left| \sum_{i=1}^{2N} W_i f(Z_i) \right|, \tag{21}$$

where $\mathcal{L}_{\mathcal{X}}^1$ is the class of real-valued 1-Lipschitz maps on $\mathcal{X}$. Also, $W_i = \frac{1}{N}$, $Z_i = Y_i$ for $i = 1, \cdots, N$ and $W_{N+i} = -\frac{1}{N}$, $Z_{N+i} = X_i$ for $i = 1, \cdots, N$. The solution to (21) can be achieved by solving a linear program (Sriperumbudur et al., 2012). As such, we suffer at the most a deterministic error due to a particular choice of augmentation. Now, combining the results obtained so far and taking expectation, we get

$$\mathbb{E}_T \mathcal{W}_1(\Psi_\Omega^\sigma(\hat{\mathbb{P}}_{T,N}), \Psi_\Omega^\sigma(\mathbb{P})) - \gamma_N \leq \mathcal{O}(M^{-\frac{\pi}{d}}) + \mathcal{O}(d^2 N^{-\frac{1}{\max\{d,2\}}}),$$

where $\gamma_N = \Lambda \mathbb{E}_{X,T} \left[ \sup_{f \in \mathcal{L}_{\mathcal{X}}^1} \left| \sum_{i=1}^{2N} W_i f(Z_i) \right| \right]$. Hence the proof.

## B.2 Proof of Theorem 3.6

The proof follows as a corollary to Raginsky et al. (2017) that we reiterate with additionally required context for a smooth read. To put our reparametrized formulation into perspective, let us first define

$$\mathcal{G}(\mathbf{x}, U_\Omega) = \frac{1}{n} \sum_{j=1}^{n} \nabla \mathbb{P}_{\Omega_{I_j}}(\mathbf{x}),$$

where $U_\Omega = (\Omega_{I_1}, \cdots, \Omega_{I_n})$ given that $I_1, \cdots, I_n$ are random permutations of $1, \cdots, n$. Also, $\mathcal{F}(\mathbf{x}) = \int \mathbb{P}_\Omega(\mathbf{x}) \mu(d\Omega)$, where $\mu$ denotes the law $\Omega$ follows such that $\mathcal{F}^* := \min_{\mathbf{x} \in \mathcal{X}} \mathcal{F}(\mathbf{x})$. The corresponding empirical risk at a realized set of parametric values is given as $\mathcal{F}_\Omega(\mathbf{x}) = \frac{1}{n} \sum_{i=1}^{n} \mathbb{P}_{\Omega_i}(\mathbf{x})$. Under this setup, our update rules, conditional on a given value of $\Omega$, corresponding to SGHMC, boil down to

$$\mathbf{p}_{j+1} = (1 - \delta_1)\mathbf{p}_j - \delta_2 \mathcal{G}(X_j, U_{\Omega,j})|_{X_j = \mathbf{s}_j} + \delta_3 \mathbf{r}_{j+1},$$
$$\mathbf{s}_{j+1} = \mathbf{s}_j + \delta_2 \mathbf{p}_j,$$

where the initializations $\mathbf{p}_0, \mathbf{s}_0$ and $\{U_{\Omega,j}\}_{j \in \mathbb{N}}$, $\{\mathbf{r}_j\}_{j \in \mathbb{N}}$ are mutually independent.

Now, let $\mathbf{s}_j$ be the output after $j \in \mathbb{N}$ iterations, and $(\hat{\mathbf{s}}^*, \hat{\mathbf{p}}^*)$ are instances that follow $\mathcal{D}(\hat{\mathbf{s}}^*, \hat{\mathbf{p}}^* | \Omega) = \Pi_\Omega$, namely the Gibbs distribution as defined earlier. Fragmenting the excess risk realized at $j$ into distinct sources of variation yields

$$\mathbb{E}(\mathcal{F}(\mathbf{s}_j)) - \mathcal{F}^* = \underbrace{\mathbb{E}(\mathcal{F}(\mathbf{s}_j)) - \mathbb{E}(\mathcal{F}(\hat{\mathbf{s}}^*))}_{\mathcal{E}_1} + \underbrace{\mathbb{E}(\mathcal{F}(\hat{\mathbf{s}}^*)) - \mathbb{E}(\mathcal{F}_\Omega(\hat{\mathbf{s}}^*))}_{\mathcal{E}_2} + \underbrace{\mathbb{E}(\mathcal{F}_\Omega(\hat{\mathbf{s}}^*)) - \mathcal{F}^*}_{\mathcal{E}_3}, \tag{22}$$

where $\mathcal{E}_1$ denotes the excess population risk due to SGHMC, taking Gibbs algorithm as the baseline. The terms $\mathcal{E}_2, \mathcal{E}_3$ respectively signify the generalization error associated with the Gibbs algorithm and the extent of expected suboptimality due to the same.

Let us concentrate on $\mathcal{E}_3$ first. Given that $\mathcal{F}^*$ is achieved at the value $\mathbf{x}^*$,

$$\mathbb{E}(\mathcal{F}_\Omega(\hat{\mathbf{s}}^*)) - \mathcal{F}^* = \mathbb{E}\left[\mathcal{F}_\Omega(\hat{\mathbf{s}}^*) - \min_{\mathbf{x} \in \mathcal{X}} \mathcal{F}_\Omega(\mathbf{x})\right] + \mathbb{E}\left[\min_{\mathbf{x} \in \mathcal{X}} \mathcal{F}_\Omega(\mathbf{x}) - \mathcal{F}_\Omega(\mathbf{x}^*)\right]$$
$$\leq \mathbb{E}\left[\mathcal{F}_\Omega(\hat{\mathbf{s}}^*) - \min_{\mathbf{x} \in \mathcal{X}} \mathcal{F}_\Omega(\mathbf{x})\right]$$
$$\leq \delta' \log\left(\frac{e\Delta}{m}\left(\frac{v}{2\delta'} + 1\right)\right),$$

where $\delta' = \frac{d\delta_3^2}{4\delta_1}$. The last inequality is due to Proposition 11 in Raginsky et al. (2017).

To upper bound $\mathcal{E}_2$, observe that

$$\mathbb{E}(\mathcal{F}(\hat{\mathbf{s}}^*)) - \mathbb{E}(\mathcal{F}_\Omega(\hat{\mathbf{s}}^*)) = \mathbb{E}\left[\mathcal{F}_{\Omega'}(\hat{\mathbf{s}}^*) - \mathcal{F}_\Omega(\hat{\mathbf{s}}^*)\right]$$
$$= \frac{1}{n} \sum_{i=1}^{n} \mathbb{E}\left[\mathbb{P}_{\Omega_i'}(\hat{\mathbf{s}}^*) - \mathbb{P}_{\Omega_i}(\hat{\mathbf{s}}^*)\right],$$

where $\Omega' = \{\Omega_i'\}_{i=1}^{n} \sim \mu^{\otimes n}$ are i.i.d. replicates drawn independently of $\Omega$ and $\hat{\mathbf{s}}^*$. Now, expanding the term inside the summation we get

$$\mathbb{E}\left[\mathbb{P}_{\Omega_i'}(\hat{\mathbf{s}}^*) - \mathbb{P}_{\Omega_i}(\hat{\mathbf{s}}^*)\right] = \int \mu^{\otimes n}(d\Omega) \int \mu(d\Omega_i') \int \pi_\Omega(d\mathbf{s})\left[\mathbb{P}_{\Omega_i'}(\mathbf{s}) - \mathbb{P}_{\Omega_i}(\mathbf{s})\right]$$
$$= \int \mu^{\otimes n}(d\Omega_1, \cdots, d\Omega_i', \cdots, d\Omega_n) \int \mu(d\Omega_i) \int \pi_{(\Omega_1, \cdots, \Omega_i', \cdots, \Omega_n)}(d\mathbf{s}) \mathbb{P}_{\Omega_i}(\mathbf{s})$$
$$- \int \mu^{\otimes n}(d\Omega_1, \cdots, d\Omega_i, \cdots, d\Omega_n) \int \mu(d\Omega_i') \int \pi_{(\Omega_1, \cdots, \Omega_i, \cdots, \Omega_n)}(d\mathbf{s}) \mathbb{P}_{\Omega_i}(\mathbf{s})$$
$$= \int \mu^{\otimes n}(d\Omega) \int \mu(d\Omega_i') \left(\int \pi_{(\Omega_1, \cdots, \Omega_i', \cdots, \Omega_n)}(d\mathbf{s}) \mathbb{P}_{\Omega_i}(\mathbf{s}) - \int \pi_\Omega(d\mathbf{s}) \mathbb{P}_{\Omega_i}(\mathbf{s})\right). \tag{23}$$

Now, applying Proposition 12 of Raginsky et al. (2017) on (23) we obtain an upper bound on $\mathcal{E}_2$ given as

$$\mathcal{E}_2 \lesssim \frac{2d}{\delta' n}\left[\Delta^2\left(\frac{v+2\delta'}{m}\right)+B^2\right],$$

where $B \geq 0$ is such that $||\nabla\mathbb{P}_\Omega(0)|| \leq B$ and the suppressed constant is the one associated with the logarithmic Sobolev inequality followed by $\Pi_\Omega$ (Raginsky et al. (2017), Proposition 9).

To obtain an upper bound for the error $\mathcal{E}_1$, first, observe that by Assumption 3.2, for any given $\Omega$, $||\nabla\mathbb{P}_\Omega(\mathbf{x})|| \leq \Delta||\mathbf{x}|| + B, \forall\mathbf{x} \in \mathcal{X}$ (Raginsky et al. (2017), Lemma 2). Denoting the marginal distribution of $\mathbf{s}_j$, given $\Omega$, by $\varrho_j := \mathcal{D}(\mathbf{s}_j)$ we can write

$$\mathbb{E}(\mathcal{F}(\mathbf{s}_j)) - \mathbb{E}(\mathcal{F}(\hat{\mathbf{s}}^*)) = \int \mu^{\otimes n}(d\Omega)\left[\int\mathcal{F}(\mathbf{s})\varrho_j(d\mathbf{s}) - \int\mathcal{F}(\mathbf{s})\pi_\Omega(d\mathbf{s})\right].$$

The term on the write possesses a natural upper bound based on the following lemma.

**Lemma B.1** (Raginsky et al. (2017), Lemma 6; Chau & Rásonyi (2022), Lemma 5.2). *Let $\alpha, \beta \in \mathcal{P}^2(\mathcal{X})$ (distributions having finite second moments) and $h : \mathcal{X} \to \mathbb{R}$ such that $h \in C^1$ satisfies*

$$||\nabla h(\mathbf{x})|| \leq \kappa_1||\mathbf{x}|| + \kappa_2, \quad \forall\mathbf{x} \in \mathcal{X}$$

*where $\kappa_1 > 0$ and $\kappa_2 \geq 0$. Then we have*

$$\left|\int_{\mathcal{X}} h\, d\alpha - \int_{\mathcal{X}} h\, d\beta\right| \leq (\kappa_1\sigma + \kappa_2)\mathcal{W}_2(\alpha, \beta),$$

*where $\sigma^2 = \int\alpha(d\mathbf{x})||\mathbf{x}||^2 \vee \int\beta(d\mathbf{x})||\mathbf{x}||^2$.*

Hence, in our case

$$\left|\int\mathcal{F}(\mathbf{s})\varrho_j(d\mathbf{s}) - \int\mathcal{F}(\mathbf{s})\pi_\Omega(d\mathbf{s})\right| \leq (\Delta\sigma + B)\mathcal{W}_2(\mathcal{D}(\mathbf{s}_j), \pi_\Omega),$$

given $\sigma = \left[\int\varrho_j(d\mathbf{s})||\mathbf{s}||^2\right]^{\frac{1}{2}} \vee \left[\int\pi_\Omega(d\mathbf{s})||\mathbf{s}||^2\right]^{\frac{1}{2}} < \infty$. Moreover, Theorem 2.8 and 4.1 in Chau & Rásonyi (2022) imply that there exists constants $c', K_1, K_2$ and $K_3$ that satisfy

$$\mathcal{W}_2(\mathcal{D}(\mathbf{s}_j), \pi_\Omega) \leq K_1(\delta_2^{\frac{1}{4}} + c') + K_2[\mathcal{W}_\rho(\mu_0(\mathbf{s}), \pi_\Omega)]^{\frac{1}{2}}\exp{(-jK_3\delta_2)}, \tag{24}$$

where the semi-metric based on $\rho(\cdot, \cdot)$ is defined as $\mathcal{W}_\rho(\alpha, \beta) = \inf_{\mathbf{s}\sim\alpha, \mathbf{s}'\sim\beta}\mathbb{E}[\rho(\mathbf{s}, \mathbf{s}')]$, and $\mu_0(\mathbf{s})$ is the $\mathbf{s}$-marginal of the initial state distribution. As such,

$$\mathcal{E}_1 \leq (\Delta\sigma + B)\left[K_1(\delta_2^{\frac{1}{4}} + c') + K_2[\mathcal{W}_\rho(\mu_0(\mathbf{s}), \pi_\Omega)]^{\frac{1}{2}}\exp{(-jK_3\delta_2)}\right].$$

To show that the term on the right of (24), and hence the error $\mathcal{E}_1$ becomes arbitrarily small, we choose $\delta_2^{\frac{1}{4}} + c' \leq \frac{1}{2K_1}\varepsilon$ and $j \geq \frac{(2K_1)^4}{K_3\varepsilon^4}\log\left(\frac{2K_2[\mathcal{W}_\rho(\mu_0(\mathbf{s}), \pi_\Omega)]^{\frac{1}{2}}}{\varepsilon}\right)$ given $\varepsilon > 0$. A simple reparametrization gives us the desired constants $c_1, c_2$ and $c_3$ in the Theorem.

## C Implementation Details

We have rigorously trained all models for 1,000 epochs, following the conventional recommendations (Tao et al., 2021; Tsai et al., 2021). We have utilized the Stochastic Gradient Descent (SGD) optimizer with a cosine learning rate scheduler, that includes a warm up for the initial 50 updates. For MoCo (He et al., 2020), BYOL (Grill et al., 2020), and NRCC, we set the base learning rate to 0.05, dynamically scaling it with the

batch size ($\beta = 0.05 \times n/256$). Notably, we have multiplied the learning rate $\beta$ for the feature extractor by 10 for the predictor networks of BYOL and NRCC, a crucial step discussed in Chen & He (2021); Grill et al. (2020) for achieving satisfactory performance.

For the other hyperparameters of NRCC, we have conducted a grid search over the possible choices of each to find the combination that worked best on average. Finally, we set the temperature $\tau$ (searched between $\{0.1, 1, 10\}$ and the regularization weight $\lambda$ (varied between $\{0.1, 0.5, 1\}$ to 0.1 each. In the case of SGHMC, we set $\delta_1$, $\delta_2$, $\delta_3$, and $\zeta$ as 0.1, 0.05, 0.99, and 1 (increasing the number of updated did not provide any considerable improvement), respectively. Following conventional guidelines, the mini-batch size was 512 for MoCo and 256 for the remaining models, including NRCC.

We have constructed the long-tailed versions of CIFAR-10 and CIFAR-20, namely CIFAR-10-LT and CIFAR-20-LT, by employing the codes from Tang et al. (2020). We have followed the guidelines of Zhou et al. (2020) and Cao et al. (2019) while constructing the LT variants of CIFAR, that resulted in two datasets both with a cluster imbalance ratio (the number of samples in the most crowded cluster divided by the number of the sample in the smallest one) of 10. Specifically, the long-tailed datasets predominantly contained samples from the majority clusters (head), while only a few samples were present in the other clusters (tail). Table 12 highlights the key properties of the datasets used in this study.

Table 12: Properties of the eight datasets used in this study.

| Datasets | Number of Sample | Number of Clusters | Resolution | Cluster Uniformity |
|---|---|---|---|---|
| CIFAR-10 (Krizhevsky & Hinton, 2009) | 60000 | 10 | $32 \times 32$ | Uniform |
| CIFAR-100 (Krizhevsky & Hinton, 2009) | 60000 | 100 | $32 \times 32$ | Uniform |
| STL-10 (Coates et al., 2011) | 13000 | 10 | $96 \times 96$ | Uniform |
| ImageNet-10 (Russakovsky et al., 2015) | 13000 | 10 | $224 \times 224$ | Uniform |
| ImageNet-Dogs (Russakovsky et al., 2015) | 19500 | 15 | $224 \times 224$ | Uniform |
| TinyImageNet (Le & Yang, 2015) | 100000 | 200 | $224 \times 224$ | Uniform |
| CIFAR-10-LT (Tang et al., 2020) | 60000 | 10 | $32 \times 32$ | Non-uniform, imbalance ratio 10 |
| CIFAR-20-LT (Tang et al., 2020) | 60000 | 20 | $32 \times 32$ | Non-uniform, imbalance ratio 10 |

## D  The Number of Iterations in SGHMC

Throughout the article, we have theoretically and empirically established how SGHMC is immensely useful for NRCC as an approximately invariant augmentation strategy. However, to effectively balance the trade-off between computational overhead and performance gain (see Table 8) we only iterate SGHMC once. This follow-up study focuses on empirically understanding the behavior of SGHMC-based augmentation when the algorithm is iterated multiple times. In Table 13 we take a path similar to that in Table 1 by considering the mean and standard deviation of Euclidean distances between 1000 negative pairs from the ImageNet-10 dataset that are generated by SGHMC over gradually increasing iterations up to 13. A closer scrutiny of Table 13 reveals that the SGHMC-augmented samples progressively converge over iterations towards a stationary benchmark as indicated theoretically (see Theorem 3.6). This is visually supported by Figure 6 that demonstrates the augmented instances for two samples (the same ones previously used in Fig 2) over progressive SGHMC iterations. In both cases, the visible dissimilarity between the augmentations corresponding to distinct samples diminishes over iterations and almost becomes identical at the end of the 11th step, signifying near convergence. From Table 13 and 1, we also observe that the negative pairs generated by SGHMC even at iteration 13 still lie at a distance $\sim 34\%$ greater compared to the maximum attainable by traditional transformations.

Table 13: Mean and Standard Deviation of Euclidean distance in the proposed BYOL+NRCC embedding space between 1000 ImageNet-10 samples and their corresponding transformed views obtained by SGHMC (ours) over progressively increasing steps.

| SGHMC iterations | Distance between negative pair in embedding space (Mean±Standard Deviation) | Difference of mean distance with the previous recorded iteration |
|---|---|---|
| Iteration 1 | 0.0112±0.0060 | - |
| Iteration 3 | 0.0092±0.0055 | -21.73% |
| Iteration 5 | 0.0071±0.0036 | -29.57% |
| Iteration 7 | 0.0060±0.0024 | -18.33% |
| Iteration 9 | 0.0055±0.0021 | -9.09% |
| Iteration 11 | 0.0053±0.0020 | -3.77% |
| Iteration 13 | 0.0051±0.0020 | -3.92% |

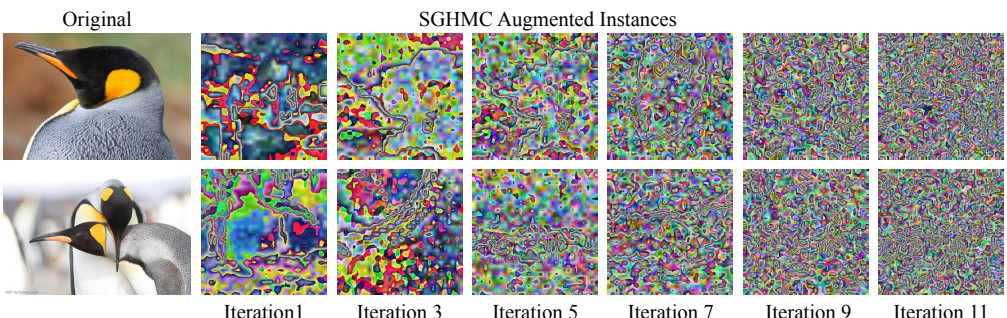

Figure 6: SGHMC augmented instances over progressively increasing iterations from 1-11 for two samples from the ImageNet-10 dataset.

