# OpenReview forum: "Enhancing Contrastive Clustering with Negative Pair-guided Regularization"
_TMLR — Accepted by TMLR_

### Review · Reviewer_wfDD · 2024-07-09

**Summary Of Contributions:**

The manuscript makes the intuitive point that in high-dimensional representation spaces, Contrastive learning (CL) methods can create deformed clusters, due to imposing the push away of two samples that are in the same cluster or lie in a local neighborhood. In addition, it is pointed out that current combinations of CL and deep clustering (DC) always require the number of clusters in advance. This motivates the proposed method: negative pair-based regularser for contrastive clustering (NRCC). The highlighted contributions are:
- New regularisation framework using hard negatives coupled with existing CL paradigms, leading to a clustering-friendly embedding space.
- Theoreical characterisation of negative pairs, facilitating cluster-friendly representation learning.
- Relieve the user from defining the number of clusters in advance.

**Audience:**

Yes

**Broader Impact Concerns:**

I do not think this is required for this paper.

**Claims And Evidence:**

Yes

**Requested Changes:**

- what are cluster--friendly embeddings formally? is there a proper definition?
- section 3.1 you mention a complex space but say that $\mathcal{X}$ is a subset of a real vector space. This is confusing.
- section 3.1: I believe your indices in $\mathcal{P}\_+$ are incorrect. The way it is written you always take the first n elements of the dataset. Perhaps you should use $x\_i\in\mathcal{N}$ in the definition of $\mathcal{P}\_+$ (or define $\mathcal{N}$ as the subset of indices of the mini-batch). The same issue holds for the othe rdefinitions in the paper (e.g. $\mathcal{P}\_{-}$). Perhaps the best solution is to talk exclusively about the mini-batch and avoid all this hassle.
- the theoretical analysis of BYOL is done in a simplified case, while one of BYOL's strengths is all the "engineering and tricks" that made it successful. I would like to see the theoretical analysis wihtout the simplification, to see if the same interpretation and reasoning hold.
- In equation 15, the $\Psi$ function you define corresponds to the $f$ function you defined in 3.1, right? Why the two notations? Why is it necessary to specify all the convolutional and fully connected layers? Can't you just say that you consider networks with bounded parameters?
- After (15) it is written "square integrable on $\mathbb{P}\times\mathbb{Q}$: what does it mean to be square integrable on a probability distribution?
- following up: what is a plug-in estimator? and what are these two estimators estimating (they are estimators of what)?
- After equation 16, you redefine $\Omega$ (that was the set of parameters) as a random variable. Please correct this. Also it is not clear what "a random variable responsible for characterizing the distribution" means.
- In assumption 3.2, you refer to $\mathbb{P}_{\Omega}$ as a function, but you defined it as a probability distribution: do you imply the pdf? Please rephrase.
- In assumption 3.4, what does the inequality with tilde sign mean? Also, what do you mean that $I_1,\ldots,I_n$ are random permutations (in plural)? Do you mean A random permutation?

**Strengths And Weaknesses:**

**Strengths**
- One of the very strong points of the manuscript is the mathematical derivations of the gradients of the modified loss, and their associated interpretation in terms of impact on the learned representations. This is of great value for the reader.
- The experimental comparison are extensive, on various datasets, using different metrics, and against several baselines. The ablation studies also show the interest of each of the proposed components.

**Weaknesses**
- The theoretical assumptions and proof for SGHMC are not intuitive (the reader is refered to Raginsky et al. (2017)), and it is unclear if the proofs in Appendix A bring something new, or they are just a mere replica of the ones in Raginsky et al. (2017). Either way, this part is very difficult to read, and the authors should provide more intuition.
- Regarding the overhead computational evaluation, we are missing the impact of the Monte-Carlo sampling: it would be interesting to report that as well.

---

> ### Author Response · Authors · 2024-07-24
> **Response to Reviewer wfDD [Part 1 of 2]**
>
> We thank the reviewer for appreciating our work and providing us with insightful comments. In the revised manuscript, we have made several updates following the recommendations. The modified portions of the article are highlighted in blue.
>
> **R1 Discussion on theoretical justification (Theorem 3.6 and Appendix A) of SGHMC in NRCC:** The paper provides a statistical theory of suitable data augmentations followed by that of SGHMC in the context of NRCC. In Theorem 3.6, we show that after a certain number of iterations, the simulated observations using SGHMC lie close to the stationary target distribution (depending on $\mathbb{P}_{\Omega}$) in distribution. This result, coupled with the empirical finding that negative pairs obtained using SGHMC on average exert greater repulsion, poses the method as an ideal augmentation technique. The proof of Theorem 3.6 however, assumes certain regularity conditions on the target distribution function and the initial iterates (originally in [1]). The assumptions are solely technical and commonly used in related literature (see, [2]-[4]). In our reparameterized setup, we reframe the assumptions according to our needs. The theorem is essentially a corollary to the discussion in [1] and not a mere replica. However, the decomposition of the excess risk being a standard technique follows exactly. Since our updation formulae are differently parameterized and we focus on proving approximate invariance, only referring to [1] seems insufficient and might make the paper difficult to follow. While we feel the proof to be self-explanatory and difficult to break down even further without making it more abstract, we include a note explaining the significance and intuition behind Theorem 3.6 in the revised manuscript.
>
> [1] Maxim Raginsky, Alexander Rakhlin, and Matus Telgarsky. Non-convex learning via stochastic gradient Langevin dynamics: a nonasymptotic analysis. Conference on Learning Theory, pp. 1674–1703. PMLR, (2017).
>
> [2] Gao, Xuefeng, Mert Gürbüzbalaban, and Lingjiong Zhu. "Global convergence of stochastic gradient Hamiltonian Monte Carlo for nonconvex stochastic optimization: Nonasymptotic performance bounds and momentum-based acceleration." Operations Research 70.5 (2022): 2931-2947.
>
> [3] Huy N Chau and Miklós Rásonyi. Stochastic gradient Hamiltonian Monte Carlo for non-convex learning. Stochastic Processes and their Applications, 149:341–368, (2022).
>
> [4] Akyildiz, O. Deniz, and Sotirios Sabanis. "Nonasymptotic analysis of Stochastic Gradient Hamiltonian Monte Carlo under local conditions for nonconvex optimization." Journal of Machine Learning Research 25.113 (2024): 1-34.
>
> **R2 Impact of Monte Carlo Sampling:** We have included a study in the second part of Section 4.5 in the revised manuscript to compare the performance and computational cost of simple Monte Carlo (MC), Hamiltonian Monte Carlo (HMC), and Stochastic Gradient Hamiltonian Monte Carlo (SGHMC) on CIFAR-10 dataset. Through this experiment, we aim to gather evidence in support of the usefulness of SGHMC in the proposed NRCC regularizer. In a nutshell, we have observed a trade-off between performance and computation among the sampling techniques. We have used SGHMC for a single iteration in the proposed NRCC framework to reach our commendable performance. In contrast, a single iteration of both simple MC and HMC, though faster, fails to perform at par with SGHMC (on average lagging 3.53% in accuracy). Whereas, increasing the number of iterations in MC and HMC puts on average a 1.63x higher computational burden compared to SGHMC.
>
> **R3 Impact of NRCC regularizer on the gradients of canonical BYOL:** At the end of Section 3.6.1 in the revised manuscript, we have included the implementation-specific heuristics used in BYOL while detailing the theoretical analysis of the impact of integrating NRCC. We have also retained the originally simplified analysis in Section 3.2 for the ease of readability and smoother flow of the discussion. In Section 3.6.1, we recall the findings of Section 3.2 and highlight how the gradients and the resultant forces are impacted when NRCC is effectively coupled with BYOL employing a stop-gradient and prediction block. Through this comparative discussion, we have also shown that the motivation in support of NRCC detailed in Section 3.2 still remains unchanged in the revised formulation in Section 3.6.1 where no simplicity assumptions are made.

---

> > ### Author Response · Authors · 2024-07-24
> > **Response to Reviewer wfDD [Part 2 of 2]**
> >
> > **R4 “cluster-friendly” embedding space:** Our notion of a “cluster-friendly” embedding follows from the work of Bo et al. [5]. This is not a strictly formal definition, but rather an attempt to intuitively characterize the intended embedding space more pedagogically. To elaborate, Bo et al. envision their embedding to be $k$-means friendly, i.e. the data points from a cluster in such a space lie around a centroid with gradually diminishing affinity, preferably revealing the convex natural cluster structure. In other words, the clusters are likely to follow spherically symmetric, bell-shaped distributions; and assuming there is no overlap, they should be linearly separable in the embedding space. In our work, we go by a generalized form of this idea and search for an embedding that attempts to conserve the modes of the original data distribution. This demands a local data density awareness from the representation learner that we aim to achieve through contrastive learning regularized by the proposed NRCC.
> >
> > [5] Yang, Bo, et al. "Towards k-means-friendly spaces: Simultaneous deep learning and clustering." International Conference on Machine Learning. PMLR, 2017.
> >
> > **R5 Notational and phrasing ambiguities in Section 3.1:**
> > A: By the word ‘complex’ we wanted to indicate the complicated nature of the support that the data distribution lies on. It often has an intrinsic structure that contains most of the variability in the data. We rephrase the sentence in the revised manuscript, making sure it does not read otherwise.
> > B: We have modified the notation for the indices signifying the entire dataset in Section 3.1. As per your suggestion, we refocus our discussion in terms of the mini-batch. In the revision, since the dataset has no specified order in the indices, the earlier problem of reappearance is avoided in the mini-batch.
> >
> > **R6 Notational ambiguities in Equation 15 against Section 3.1, along with specification of convolutional and fully connected layers:** In the introductory discussion [Section 3.1], we follow a simpler notation ($f$) for the encoding map to maintain lucidity. It is only during technical results that we resort to a rigorous definition of the transformation ($\Psi^{\sigma}_{\Omega}$) that induces $f$. The mathematical definition (15) is imperative to the discussion not only since it gives a realistic depiction of the network architecture used, but also since we exploit the universal approximation of such transforms in Proposition 3.1. Only stating “networks with bounded parameters” without the exact specifications might be incomplete and misleading to some readers.
> >
> > **R7 Square integration after Equation 15:** By the same, we mean that the associated random variable has a finite second moment with respect to $\mathbb{P} \times \mathbb{Q}$. We modify the manuscript accordingly.
> >
> > **R8 Plug-in estimator:** By ‘plug-in’, we mean the empirical distribution corresponding to $\mathbb{P}$. It can essentially be understood as the sample counterpart of the ‘population’ measure, based on a set of samples $X_{1}, X_{2}, \cdots, X_{N}$. We provide an exact definition in the revision.
> >
> > **R9 Random variable after equation 16:** The discussion in Section 3.4 starts with a general definition of the target distribution $\mathbb{P}\_{\Omega}$ that is characterized by the parameter $\Omega$. For instance, if the map $f$ is indeed induced by the network $\Psi^{\sigma}_{\Omega}$, they are exactly the set of model parameters. In general, in an SGHMC setup, $\Omega$ itself can be an instance of a random variable. For example, the model parameters are initialized typically from Gaussian distributions or borrowed from pre-trained architectures which during training go through a stochastic propagation towards an optimal value. As such, this is not a redefinition but a technical detail of broader consequence. We, however, modify the portion in the manuscript to make it clearer.
> >
> > **R10 Defining PDF as a function in Assumption 3.2:** $\mathbb{P}_{\Omega}$ signifies the target probability distribution ‘function’. Assumption 3.2 imposes a regularity condition on the same, essentially demanding that the distribution must possess a Lipschitz-smooth density. We rephrase the statement to avoid any confusion.
> >
> > **R11 Assumption 3.4 ambiguities:** We clarify the notation in the revised manuscript. Also, we do mean ‘a random permutation’ and rectify the same in the revision.

---

### Review · Reviewer_5DJk · 2024-08-02

**Summary Of Contributions:**

This submission contributes a Contrastive-Learning-based deep clustering framework named Negative pair-based Regularizer for Contrastive Clustering(NRCC), which generates cluster-friendly embeddings.

**Audience:**

Yes

**Broader Impact Concerns:**

I did not identify any concerns on the ethical implications of the work.

**Claims And Evidence:**

Yes

**Requested Changes:**

The authors have added many detailed mathematical definitions and argumentation details, which are crucial for understanding the method proposed in the article.

**Strengths And Weaknesses:**

Strengths

- The author provided a detailed mathematical process of argumentation.
- The experiments are extensive and the results are promising.
- According to what the author said in the submission, it is the first time to investigate the Stochastic Gradient Hamiltonian Monte Carlo (SGHMC) sampling method as a potential data augmentation technique in contrastive learning.

Weaknesses

- There are some writing errors in the manuscript, and the text has not been carefully proofread. e.g. in the introduction section, the last sentence of the second paragraph is repeated in the second sentence of the fourth paragraph.
- The mathematical formulas are too complex and lack necessary explanations, making the content difficult to understand. Additionally, there are some minor writing issues present in the formulation of some mathematical formulas. i.e. n Formula 7, there is a missing ‘]’ symbol and TX ≠ d X. in chapter 3.3.
- The paper only analyzes the combination of the NRCC method with InfoNCE and BYOL, which are classical contrastive learning methods. Looking forward to the results of its combination with the latest contrastive learning methods.

---

> ### Author Response · Authors · 2024-08-21
> **Response to Reviewer 5DJk [Part 1 of 1]**
>
> We thank the reviewer for the meticulous reading and the pertinent comments. The revised manuscript reflects the suggested changes and improvements. For the convenience of the reviewers, we have highlighted the modified portions in pink.
>
> **R1 Writing errors and complex mathematical formulae:** We have corrected the writing errors pointed out. In the revised version, we have also attached a table [Table 10] describing all mathematical notations used in the main paper.
>
> **R2 Coupling NRCC with other latest contrastive learning methods:** In the revised manuscript, we have added a new section 4.6 that compares the performance of NRCC coupled with Barlow Twins [6] and SimSiam [7]. From the allied Table 9, we observe that coupling NRCC failed to provide a commendable improvement over the two proposed methods, namely regularized InfoNCE and BYOL. This may be influenced by the fact that the two new baseline algorithms are not as competent as InfoNCE or BYOL given the downstream task is clustering.
>
> [6] Zbontar, Jure, et al. "Barlow twins: Self-supervised learning via redundancy reduction." International conference on machine learning. PMLR, 2021.
>
> [7] Chen, Xinlei, and Kaiming He. "Exploring simple Siamese representation learning." Proceedings of the IEEE/CVF conference on computer vision and pattern recognition. 2021.

---

### Review · Reviewer_GpT5 · 2024-08-29

**Summary Of Contributions:**

This paper demonstrates that the imbalanced learning signals from positive and negative pairs in traditional contrastive learning and clustering methods can lead to poorly differentiated embedding spaces. To address this issue, the authors propose a regularization method called Negative Pair-based Regularizer (NRCC) and a negative pair data augmentation method using Stochastic Gradient Hamiltonian Monte Carlo (SGHMC) sampling. Authors experimentally show that NRCC and SGHMC complement each other and, when combined with other methods, achieve superior performance.

**Audience:**

Yes

**Broader Impact Concerns:**

There are no specific concerns to address.

**Claims And Evidence:**

Yes

**Requested Changes:**

Could authors provide examples of consecutive images as they change over several SGHMC steps and analyze how the distances in the embedding space vary for those images?

**Strengths And Weaknesses:**

## Strengths

Most of the concerns are addressed in the experiments, and the explanations are sufficiently detailed.

1. There is extensive analysis on the effectiveness of NRCC, which strongly supports the claims made.

2. A general regularization loss term is proposed, which can be easily applied to other methods.

3. The combination of NRCC + SGHMC + GridShift shows significantly stronger performance compared to other state-of-the-art methods.

## Weaknesses

1. There is a lack of analysis on the effects of GridShift.

2. When using SGHMC, performing multiple sampling steps incurs additional time costs.

---

> ### Author Response · Authors · 2024-09-06
> **Response to Reviewer GpT5 [Part 1 of 1]**
>
> We thank the reviewer for the insightful reviews. Following the suggestions, we have added a new section 2.3 under Related Works to discuss in detail GridShift and its role in the current work. We have also performed an analysis of the augmentation as SGHMC is executed over multiple iterations. The same is documented under the Appendix in Section E of the revised manuscript. The requested modifications are highlighted in dark sea green color in the updated version.
>
> **R1 There is a lack of analysis on the effects of GridShift:** The introduction of GridShift follows a long line of work addressing the problem of scalable mode-seeking clustering. In our work, we focus specifically on such methods given they do not need to know the number of clusters in advance. Moreover, the only drawback of these methods i.e. their computational burden with increasing dimension, can be alleviated by NRCC since it preserves neighborhood and emphasizes natural clusters in the dataset. NRCC embeddings can be projected on a lower-dimensional space by a neighborhood-preserving method like t-SNE or UMAP, where mode-seeking clustering can be effectively applied. Specifically, GridShift [1] shows that it maintains the linear time-complexity of its predecessor MeanShift++ [2] while achieving about 40x speedup on average. The authors showed that GridShift is theoretically equivalent to other mode-seeking clustering methods while empirically achieving a better performance. Thus, GridShift was a natural choice for us. Note that our theoretical analysis of NRCC is not dependent on the choice of the clustering algorithm. Our NRCC strategy only allows employing an additional class of methods that provides empirical and implementation benefits.
>
> [1] Kumar, Abhishek, et al. "GridShift: A Faster Mode-seeking Algorithm for Image Segmentation and Object Tracking." Proceedings of the IEEE/CVF Conference on Computer Vision and Pattern Recognition. 2022.
>
> [2] Jang, Jennifer, and Heinrich Jiang. "MeanShift++: Extremely fast mode-seeking with applications to segmentation and object tracking." Proceedings of the IEEE/CVF Conference on Computer Vision and Pattern Recognition. 2021.
>
> **R2 Additional cost for multiple iterations of SGHMC:** The reviewer is right in pointing out that multiple iterations of SGHMC are indeed a significant computational burden. Due to this, we only limit SGHMC to a single iteration that does not drastically increase the running time but provides a commendable performance boost. The experiment in Section 4.4 shows how SGHMC with a single step can achieve about 7.5% performance gain while only incurring 1.4x additional cost.
>
> **R3 SGHMC for multiple steps:** As per the suggestions, we have added the new Section E in the Appendix where Table 12 and Figure 5 extend the experiment of Section 3.5 for multiple iterations of SGHMC. The empirical findings clearly collate with the theoretical observations and further strengthen the usefulness of SGHMC in NRCC.

---

### Decision · Action_Editor_dELX · 2024-10-30

**Recommendation:** Accept with minor revision

**Comment:**

Overall, the proposed framework seems to be technically sound and moreover supported by extensive empirical validation. Most concerns raised by the reviewers are addressed in the revision with better description and additional experiments. Based on the consensus among the reviewers, I would recommend the paper to be accepted.

However, it is still necessary to clarify the main motivation and algorithm for the proposed regularization and representation learning. For example, it is unclear that why the proposed regularization losses in Eq. (5) and (6) can lead to good clustering and what is the role of an another augmentation x^c or z^c related to a hard negative. Most of all, it is hard to figure out the whole processing of the proposed representation learning. Algorithm 1, 2 and 3 in Appendix C should be located in the main body, and a figure to describe the overall processing is necessary. In addition, the performance evaluation with larger networks and datasets such as
ViTs and full ImageNet is also necessary.

**Audience:**

Contrastive representation learning for clustering would be an interesting topic, and especially the hard negative mining based on the data augmentation and the mode-seeking clustering can receive a lot of interest from the community including TMLR's audience.

**Claims And Evidence:**

This paper proposes a new regularization loss and data augmentation technique for leveraging hard negatives in contrastive representation learning targeting for clustering. Especially, the proposed framework, negative pair-based regularizer for contrastive clustering (NRCC), can retain proper cluster structures in a lower-dimensional space, which allows to use a mode-seeking clustering algorithm without requiring a predetermined number of clusters. Overall, the proposed framework seems to be technically sound and moreover supported by extensive empirical validation.

---

> ### Author Response · Authors · 2024-11-27
> **Response to Action Editor dELX [Part 1 of 1]**
>
> First and foremost, let us thank the reviewing team for providing rigorous and extremely helpful reviews. The review process significantly aided us in improving the quality of the manuscript and enabled a better validation of our work. In the current camera ready copy we are adding the following prescribed changes.
> 1. Algorithm 1,2, and 3 are moved from Appendix C to the main paper at their respective best fit places.
> 2. A new schematic in Figure 3 in the updated manuscript provides a view of the entire workflow.
> 3. New experiments on ImageNet-1k and Visual Transformers are included in Section 4.7 of the revised manuscript.
> 4. We have thoroughly scrutinised the updated manuscript to weed out typographical errors. We have further made some rephrasing and rewriting to improve the quality and clarity.
> 5. Necessary steps for preparing the camera ready copy are taken, for example removing review-phase annotations, lifting anonymity, adding publication details, etc.
> 6. The final codebase is made publicly available while the link of the same is provided in the revised manuscript.
> Thank you again for reviewing and accepting our work.